# Precise synthesis and photovoltaic properties of giant molecule acceptors

Hongmei Zhuo[1,2,6], Xiaojun Li [1,2,6] ✉, Jinyuan Zhang[1,6], Can Zhu[1,2], Haozhe He[1,2], Kan Ding[3], Jing Li[4], Lei Meng [1,2], Harald Ade [3] ✉ & Yongfang Li [1,2,5] ✉

Series of giant molecule acceptors DY, TY and QY with two, three and four small molecule acceptor subunits are synthesized by a stepwise synthetic method and used for systematically investigating the influence of subunit numbers on the structure-property relationship from small molecule acceptor YDT to giant molecule acceptors and to polymerized small molecule acceptor PY-IT. Among these acceptors-based devices, the TY-based film shows proper donor/acceptor phase separation, higher charge transfer state yield and longer charge transfer state lifetime. Combining with the highest electron mobility, more efficient exciton dissociation and lower charge carrier recombination properties, the TY-based device exhibits the highest power conversion efficiency of 16.32%. These results indicate that the subunit number in these acceptors has significant influence on their photovoltaic properties. This stepwise synthetic method of giant molecule acceptors will be beneficial to diversify their structures and promote their applications in high-efficiency and stable organic solar cells.

Organic solar cells (OSCs) have attracted remarkable interests due to their advantages in manufacturing flexible and wearable photovoltaic devices through low-cost solution processing methods[1–3]. Among the OSCs, all-polymer solar cells (all-PSCs) with a *p*-type conjugated polymer as donor and an *n*-type conjugated polymer as acceptor, are promising for the fabrication of flexible devices due to their good flexibility and morphology stability[4,5]. However, the early polymer acceptors used in all-PSCs exhibited weak absorbance in the near-infrared region and unsuitable molecular packing[6,7], which limited the further development of all-PSCs. To address these issues and increase the power conversion efficiency (PCE) of all-PSCs, Zhang and Li et al.[8] proposed the concept of polymerizing small molecule acceptor (PSMA) to construct the emerging generation polymer acceptors with the narrow bandgap small molecule acceptors (SMAs) as the key building blocks[9,10]. The PSMAs possess the advantages of its SMA

building blocks with low bandgap, strong absorption in the near-infrared region, suitable molecular packing and smaller exciton binding energy than the SMA building block, which prompt the PCE of the all-PSCs to over 17% recently[11–14]. In addition, from SMAs to PSMAs, the intrinsically higher glass transition temperature ($T_g$) and lower diffusion property of the PSMAs with large molecular weight significantly increase the morphology stability of the PSMAs-based OSCs[15].

Although remarkable achievements have been made by PSMAs in the development of all-PSCs, the photovoltaic performance of PSMAs is largely dependent on their molecular weight with batch variation[16–18]. To circumvent the batch variance of PSMAs and achieve lower diffusion properties simultaneously, innovative materials with precisely defined structure and high molecular weight close to polymer need to be developed, while this will be challenging for their synthesis. Fortunately, the structural characteristics of PSMAs provide the possibility

[1]Beijing National Laboratory for Molecular Sciences, CAS Key Laboratory of Organic Solids, Institute of Chemistry, Chinese Academy of Sciences, Beijing 100190, China. [2]School of Chemical Science, University of Chinese Academy of Sciences, Beijing 100049, China. [3]Department of Physics and Organic and Carbon Electronics Laboratories (ORaCEL), North Carolina State University, Raleigh, NC 27695, USA. [4]Key Laboratory of Photochemical Conversion and Optoelectronic Materials, Technical Institute of Physics and Chemistry, Chinese Academy of Sciences, Beijing 100190, China. [5]Laboratory of Advanced Optoelectronic Materials, College of Chemistry, Chemical Engineering and Materials Science, Soochow University, Suzhou, Jiangsu 215123, China. [6]These authors contributed equally: Hongmei Zhuo, Xiaojun Li, Jinyuan Zhang. ✉e-mail: lixiaojun@iccas.ac.cn; hwade@ncsu.edu; liyf@iccas.ac.cn

to solve this problem. Firstly, the currently reported high performance PSMAs exhibit relatively low molecular weight ($M_n < 10000$)[17,19], and too large molecular weight may lead to severe entanglement of polymer chains, thus resulting in unsuitable phase morphology in the active layer. Secondly, the SMA building blocks have relatively large molecular structures, so the connection of several SMA units can conveniently obtain molecules with molecular weights of several thousands. Therefore, by linking multiple SMA subunits through the bridge units, the acceptors with well-defined structures and molecular weights close to that of high performance PSMAs can be prepared. Such kind of acceptors can be defined as giant molecule acceptors (GMAs)[20], which is composed of SMA subunits connected by conjugated linking unit, and the GMAs possess the advantages of PSMAs with higher stability and SMAs with fixed molecular weight[20,21].

Recently, several simple GMAs with two SMA subunits have been constructed[22–24], and the complex GMAs containing more than two SMA subunits, such as OY3 and OY4, were isolated from polymerization reaction[25]. Therefore, the lack of practical synthetic methods and low structural diversity of the complex GMAs are the key points to be solved for the development of GMAs. Meanwhile, there is still lack of systematic research on the influence of the structure and number of SMA subunits on the photovoltaic performance of the GMAs.

Here, based on the SMA subunit of YDT in the representative PSMA PY-IT, a series of GMAs including DY (with two YDT subunits), TY (with three YDT subunits) and QY (with four YDT subunits, which possesses the molecular weight close to that of the PY-IT) were designed for systematically investigating the influence of the number of the SMA subunits on the physicochemical and photovoltaic properties of the GMAs. Meanwhile, a key link is provided by these GMAs through gradually increasing their molecular weight to in-depth understand the performance differences from SMAs to PSMAs. For the synthesis of the target GMAs, retrosynthetic analysis was used to select suitable synthons for the synthesis route. Through trials, we finally found the synthetic method of successive module superposition, and the GMAs (DY, TY and QY) with YDT as the SMA subunit and thiophene as linking unit were precisely synthesized, based on the boron trifluoride etherate-catalyzed Knoevenagel condensation and the classic Stille cross-coupling reaction. Besides, the corresponding SMA YDT and PSMA PY-IT were also synthesized for convenient comparison (Fig. 1). From YDT to DY, TY, QY and to PY-IT, the increased molecular weight of these materials enables the gradually red-shifted UV-vis absorption spectra, narrowed electrical bandgaps, decreased molecular diffusion and increased delocalization of excitons within their molecular chains. Interestingly, the TY-based device exhibits more efficient exciton dissociation, higher charge transportation, and lower carrier recombination properties than the other devices based on these acceptors. Besides, the TY and QY-based blend films also exhibit a higher CT state yield and longer CT state lifetime due to their proper phase size to balance the CT and charge recombination, comparing to the slower charge transfer in the PY-IT based blend film for its oversized phase domain and the rapid decay of CT state to ground state in the YDT based blend film owing to its small phase separation. Finally, the TY-based device demonstrated the highest PCE of 16.32%, followed by the PCEs of 15.47%, 14.88%, 14.97% and 7.47% for the QY, DY, PY-IT and YDT -based devices respectively. These results indicate that the number of SMA subunits in GMAs have significant influence on their photovoltaic performances and device stability, and the structural model built by the GMAs based on gradually increasing their molecular weight is of great significance for the in-depth understanding the structure-performance relationship from SMAs to PSMAs.

## Results

### Synthesis and characterization
Due to the lack of synthetic methods for the GMAs containing more than two SMA units, we tried to synthesize the GMAs using a

retrosynthetic strategy, and the GMA TY with three SMA subunits was selected as an example. As shown in Supplementary Fig. 1, we designed three disconnection methods for TY. Among them, the symmetrical strategies 1 and 2 seem to be the simplest paths but we failed to obtain the target molecules by the symmetrical strategies (Supplementary Fig. 2a, b), as described in Supporting Information. Then, we tried the asymmetric disconnection strategy **3** (Supplementary Fig. 3) and we succeeded (Fig. 1). The simplified structures of GMAs with three SMA subunits (fragments I, II and III) and the two linking units (π1 and π2) are shown in Fig. 1a. Notably, the step-wisely and uniquely defined reaction sites (highlighted by the different shapes of A units) in these SMA subunits during the synthesis of the GMA, are different from the synthesis of OY3 and TYT which were separated from the products mixture of their polymerization reaction[25,26]. Our synthetic method here is a stepwise precise synthesis method, which ensures a controllable procedure to synthesize this kind of GMAs. In addition, this method enables the three fragments I, II, III and the two π-spacer linking units of the GMA to be modified independently, so that it is convenient to modify the molecular structure of the GMAs.

To investigate the structure-performance relationship from SMA to GMAs and to PSMA, the same SMA subunit YDT and thiophene π-spacer that make up PY-IT were chosen to construct the GMAs DY, TY and QY. Therefore, the above-mentioned fragments I, II and III are unified as YDT, and the π-spacers are unified as thiophene π-spacers. The specific synthetic routes of these materials are shown in Fig. 1b, in which the SMA subunit YDT and the corresponding PSMA PY-IT were prepared simultaneously during the synthesis of GMAs. The boron trifluoride etherate-catalyzed Knoevenagel condensation and the classic Stille cross-coupling reaction are the mainstays throughout these synthetic routes. YDT was synthesized directly by Knoevenagel condensation of BTCHO with the end groups IC and IC-Br, accompanying with the production of the double IC-Br end groups substituted **m1** and the asymmetric IC and IC-Br substituted **m2** in one-pot reaction. Subsequently, the PSMA PY-IT can be obtained by coupling **m1** with 2,5-bis(trimethylstannyl)thiophene. For the synthesis of GMAs DY, TY and QY, the asymmetric monobromated **m3** was synthesized by Knoevenagel condensation of BTCHO and end group IC-Br. Then coupling **m3** with the asymmetric molecule **m2** and 2,5-bis(trimethylstannyl)thiophene yields the binuclear mono-aldehyde compound **m4**. Subsequently, **m5** was obtained through Knoevenagel condensation between **m4** and end group IC-Br in high yields. It is important to emphasize that the Knoevenagel condensation employed here should be catalyzed by boron trifluoride etherate[27]. Compared with the traditional base-catalyzed (like pyridine) conditions, this reaction condition can effectively prevent the reversible reaction during the terminal substitution and avoid the generation of by-products (Supplementary Fig. 4). Meanwhile, this method can effectively improve the reaction efficiency with shortened reaction time and increased yield. Finally, coupling **m5** with 2,5-bis(trimethylstannyl)thiophene and **m2**, the target products GMAs DY, TY and QY were obtained in one-pot.

The detailed synthesis processes of the GMAs are described in Supporting Information. The chemical structures mentioned above were confirmed by $^1H$ and $^{13}C$ NMR spectra (Supplementary Figs. 5–20) and matrix-assisted laser desorption ionization time of flight mass spectrometry (MALDI-TOF MS) (Supplementary Figs. 21–28). The gradually increased molecular weight of SMA YDT and GMAs DY, TY and QY are 1827.07, 3734.13, 5642.19 and 7550.25, respectively. The number average molecular weight ($M_n$) of PY-IT is 8210 Daltons with a polydispersity index (PDI) of 2.08, according to the high-temperature gel permeation chromatography (GPC) result (Supplementary Fig. 29). This is consistent with the molecular weight of most of the reported high-performance PSMAs based on the A-DA'D-A structured SMAs[28,29]. All the materials exhibit good solubility in chloroform, ensuring the processability of corresponding photovoltaic devices in chloroform. Furthermore, the thermal stability of these materials was investigated

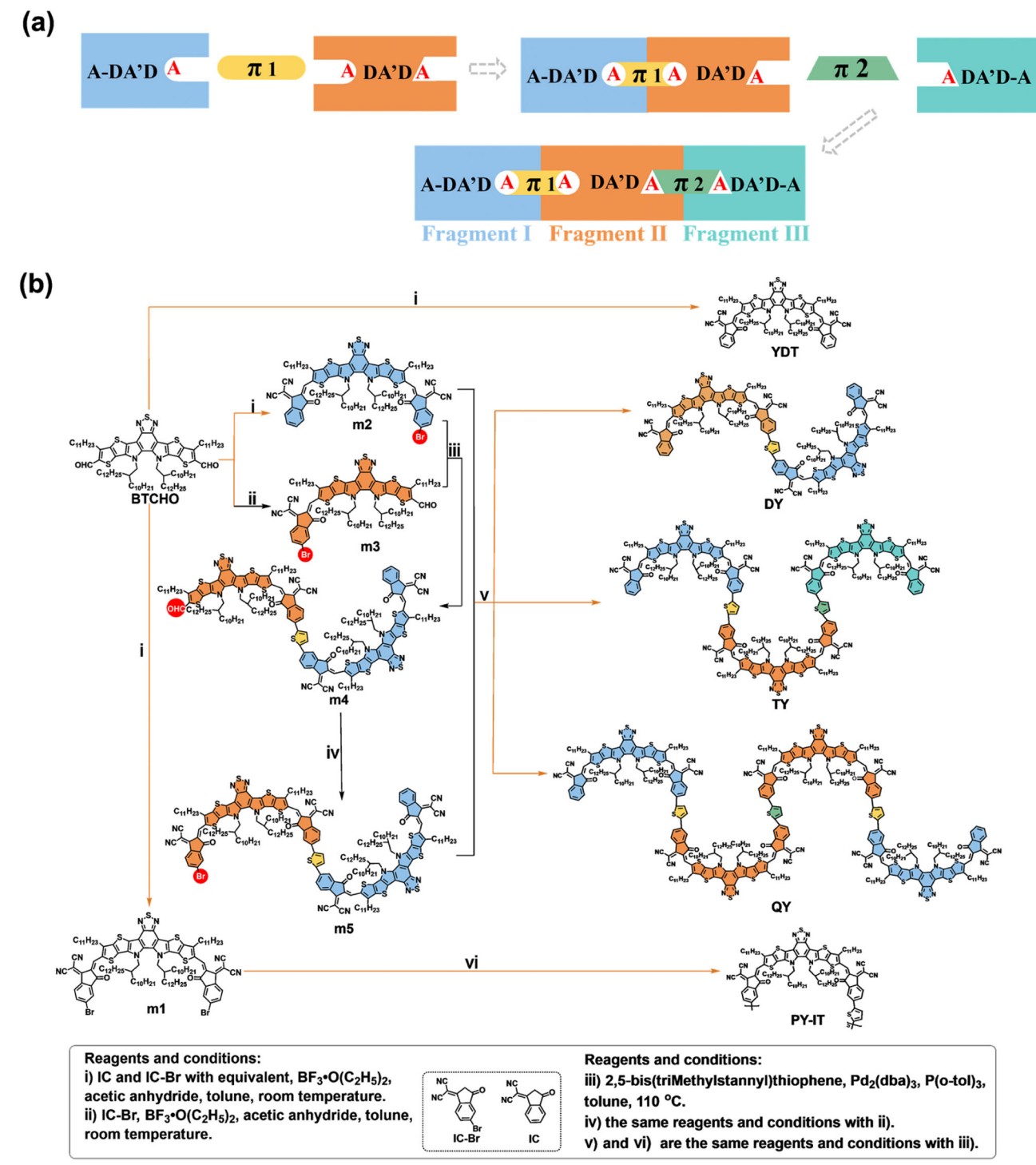

**Fig. 1 | Materials design and synthesis of giant molecule acceptors. a** The simplified diagram of the GMAs with three SMA subunits. **b** The synthetic routes of YDT, DY, TY, QY and PY-IT. (The blue, orange and cyan color-filled parts represent the independent subunit and the yellow and green color-filled parts represent the π-spacer linking units. The red circles represent the reaction sites. The Roman numerals i-vi represent the synthetic steps with the reaction conditions described in the bottom of the figure.).

by thermogravimetric analysis (TGA), as shown in Supplementary Fig. 30. The 5% weight-loss temperature for YDT, DY, TY, QY and PY-IT are 321 °C, 327 °C, 328 °C, 335 °C and 355 °C respectively under nitrogen atmosphere, the gradually increased thermal decomposition temperature of these acceptors indicates that the larger molecular size contributes to better thermal stability of these materials.

UV-vis absorption spectra of these acceptors in chloroform solutions and solid films were measured and the results were

exhibited in Fig. 2a, b. The detailed parameters were summarized in Table 1. The absorption peak wavelength ($\lambda_{max}$) of these materials in solutions are 720 nm for YDT, 735 nm, 746 nm and 762 nm for DY, TY and QY respectively, 796 nm for PY-IT, which are gradually redshifted with the increased molecular size from SMA to GMAs and further to PSMA. Besides, the temperature-dependent UV-vis absorption spectra of these acceptors were also measured in the temperature range from 60 °C to 20 °C in chloroform to investigate

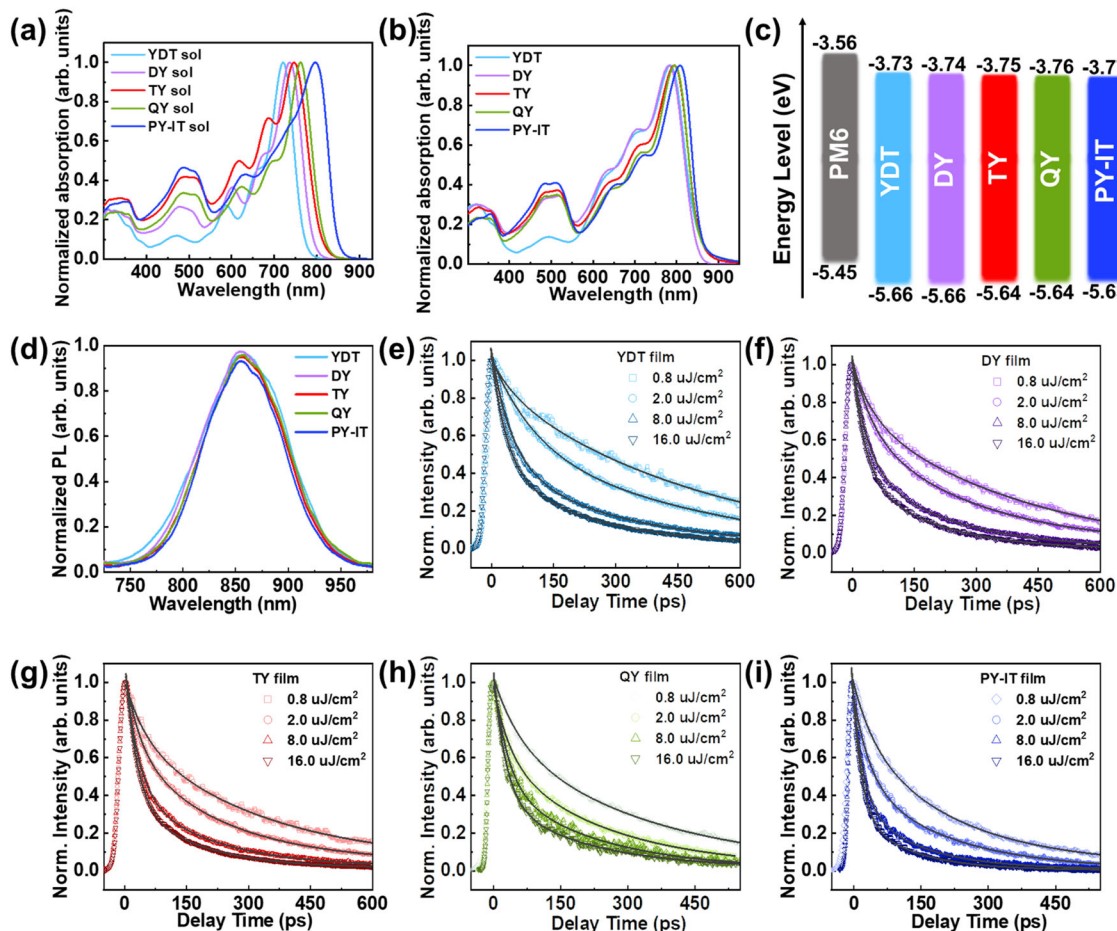

**Fig. 2 | Materials characterization of acceptors.** Normalized UV-vis absorption spectra of the acceptors of YDT, DY, TY, QY and PY-IT in (**a**) solutions and (**b**) neat films respectively. **c** Energy alignment of the materials in OSCs; **d** steady-state photoluminescence spectra of the acceptors. **e–i** Photoluminescence decays of the acceptor films of YDT, DY, TY, QY and PY-IT for a range of excitation fluences. Source data are provided as a Source Data file.

**Table 1 | Summary of optical properties and electronic energy levels, exciton lifetime τ, annihilation rate constant γ, annihilation radius R, exciton diffusion coefficient D and exciton diffusion length $L_D$ of YDT, DY, TY QY and PY-IT**

| Acceptor | $\lambda_{max}^{sol}$ (nm) | $\lambda_{max}^{film}$ (nm) | $\varepsilon_{max}^{film}$ ($10^5$ cm$^{-1}$) | HOMO (eV) | LUMO (eV) | τ (ns) | γ ($10^{-8}$ cm$^3$s$^{-1}$) | R (nm) | D ($10^{-3}$ cm$^2$s$^{-1}$) | $L_D$ (nm) |
|---|---|---|---|---|---|---|---|---|---|---|
| YDT | 720 | 785 | 1.02 | −5.66 | −3.73 | 1.05 ± 0.08 | 0.88 ± 0.01 | 2.36 | 1.48 ± 0.02 | 12.5 ± 0.5 |
| DY | 735 | 781 | 1.21 | −5.66 | −3.74 | 1.71 ± 0.09 | 1.1 ± 0.01 | 2.05 | 2.14 ± 0.02 | 19.1 ± 0.6 |
| TY | 746 | 793 | 1.42 | −5.64 | −3.75 | 2.70 ± 0.15 | 1.5 ± 0.03 | 2.08 | 2.87 ± 0.06 | 27.8 ± 1.1 |
| QY | 762 | 795 | 1.36 | −5.64 | −3.76 | 2.71 ± 0.10 | 1.7 ± 0.03 | 1.69 | 4.00 ± 0.07 | 32.9 ± 0.9 |
| PY-IT | 796 | 808 | 1.44 | −5.63 | −3.77 | 2.23 ± 0.08 | 2.1 ± 0.05 | 1.62 | 5.40 ± 0.12 | 34.5 ± 1.2 |

their aggregation properties in solutions (Supplementary Fig. 31). All these acceptors show similar spectral change behavior with their increased and red-shifted maximum absorption peaks from 60 °C to 20 °C, which indicates the similar molecular interactions and aggregation behavior of these acceptors in chloroform solutions. From solution to film, the absorption of these molecules shows different degrees of redshift. In details, the film $\lambda_{max}$ is 785 nm, 781 nm, 793 nm, 795 nm, and 808 nm for YDT, DY, TY, QY and PY-IT respectively. Among them, the $\lambda_{max}$ of SMA YDT shows the largest redshift of 65 nm from solution to film, comparing to those of medium redshift for GMAs (46 nm, 47 nm and 33 nm for DY, TY and QY respectively) and minimum redshift for PY-IT (12 nm). The gradually decreased absorption redshift from SMA to PSMA indicates their gradually weakened molecular aggregation characteristics, which is mainly due to the restricted molecular accumulation for the increased molecular size. Except for the $\lambda_{max}$ mentioned above, there are other three absorption peaks at ~710 nm, ~640 nm and ~510 nm. The main peak and the shoulder peak at ~710 nm of the corresponding acceptors should be ascribed to the electronic transition of S0→S1 with partial charge transfer (CT) character and its first vibronic (0–1) side band, respectively[25]. The shoulder peak at ~640 nm is a separate, weakly allowed electronic transition instead of a vibronic side band[30]. Furthermore, the pronounced absorption band at ~510 nm for the three GMAs and PY-IT is assigned to the π-π* transition peak, which is attributed to the well-delocalized lowest unoccupied molecular orbitals (LUMOs) that extends from the SMA subunit to the thiophene π-spacer for the γ-linking site on IC end group[19,31]. In addition, the film extinction coefficients ($\varepsilon_{max}$) were measured to be 1.02, 1.21, 1.42, 1.36 and 1.44 × $10^5$ cm$^{-1}$ for YDT, DY, TY, QY and PY-IT, respectively (Table 1). The molecular $\varepsilon_{max}$ gradually increases with the increased molecular size and remains at high level after the molecular backbone contains more than three

SMA subunits (GMA TY). The higher $\varepsilon_{max}$ may enhance the light harvest of the corresponding film, contributing to higher $J_{sc}$ in OSCs.

The highest occupied molecular orbital (HOMO) and the LUMO energy levels ($E_{HOMO}/E_{LUMO}$) of these materials were estimated from the onset oxidation/reduction potentials ($\varphi_{ox}/\varphi_{red}$) in the electrochemical cyclic voltammograms (Supplementary Fig. 32)[32]. The corresponding energy level diagrams and the detailed parameters are shown in Fig. 2c and Table 1. The $E_{HOMO}/E_{LUMO}$ values for YDT, DY, TY, QY and PY-IT are calculated to be −5.66/−3.73 eV, −5.66/−3.74 eV, −5.64/−3.75 eV, −5.64/ −3.76 eV and −5.63/−3.77 eV, respectively. The gradually down-shifted $E_{LUMO}$ and the slightly up-shifted $E_{HOMO}$ correspond to the narrowed energy bandgap from YDT to DY, TY, QY and to PY-IT.

In the OSCs, the active layer absorbs photons to produce excitons (bounded electron-hole pairs). The exciton dissociation occurs at the D/A interface in the active layer, thus the exciton diffusion property is crucial for the photovoltaic donor and acceptor materials, because the excitons generated in the donor or acceptor domains need to reach the D/A interface before decaying to ground state. To explore how the number of SMA subunits influences the exciton diffusion properties of the acceptors, time-resolved photoluminescence experiments (TRPL) were carried out for the acceptors, and singlet-singlet annihilation (SSA) method was used to measure exciton diffusion parameters[33,34]. The five acceptor films display similar PL spectra profiles as shown in Fig. 2d, while their PL decay dynamics are different. Figure 2e–i exhibit that the PL decay rate of the acceptors substantially raises as the pump fluence is increased from 0.8 uJ/cm² to 16 uJ/cm², which is a clear sign of singlet-singlet annihilation. Moreover, it is obvious that under the same excitation fluence, the PL decay rate accelerates as the number of acceptor subunits increases. The excitons in PY-IT show the fastest decay rate, indicating strong exciton annihilation in the film[35]. The PL decay with SSA is described by the rate equation[36]:

$$\frac{dn(t)}{dt} = -kn(t) - \frac{1}{2}\gamma n^2(t) \tag{1}$$

where $n(t)$ is the total initial exciton population, $k$ is the intrinsic decay rate constant free of annihilation and $\gamma$ the annihilation rate constant. In order to determine $k$, we measured the PL decays using solutions at very low concentration where PL decay is fluence-independent (Supplementary Fig. 33). The exciton decays at various excitation densities were fitted by using Eq. (1) to obtain the annihilation rate constant $\gamma$, details were provided in method section. An apparent trend is observed that as the SMA subunits increase from YDT to PY-IT, the annihilation rate constant increases, resulting in fast depletion of the excitons. Then the rate constant $\gamma$ is used to determine the exciton diffusion coefficient $D$ using[37]:

$$D = \frac{\gamma}{8\pi R} \tag{2}$$

Where R is the annihilation radius from the $d_{100}$ spacing in GIWAXS data. The one-dimensional exciton diffusion length can be calculated by:

$$L_D = \sqrt{D\tau} \tag{3}$$

$\tau$ is the intrinsic exciton lifetime extracted from dilute acceptor samples and the results are listed in Table 1. The $L_D$ values of YDT, DY, TY, QY and PY-IT are 12.5 nm, 19.1 nm, 27.8 nm, 32.9 nm and 34.5 nm, respectively, indicating that the $L_D$ values increase with the increased SMA subunits of the acceptors. Although the larger molecule size may increase the intermolecular distance and leads larger static disorder of the corresponding acceptor[38], the exciton diffusion coefficient is extremely resilient to static disorder and the exciton delocalization

along π-conjugation chains somehow "redistributes" the total excitonic interactions among short- and long-range intermolecular contributions in a way that is favorable to interchain energy transfer[39]. Thus, these lead to the acceptors with elongated π-extending conjugation chains possess larger $L_D$. The enlarged $L_D$ values for the GMAs and the PSMA will benefit the excitons dissociation so that to increase the short circuit current of the corresponding OSCs.

**Photovoltaic Properties**

The OSCs with YDT, DY, TY, QY or PY-IT as acceptor and PM6 as donor were fabricated with a conventional device structure of ITO/PEDOT:PSS/PM6:acceptor /PDINN/Ag to investigate the photovoltaic performances of these acceptors. PM6 was chosen as the donor for its matched energy levels and complementary absorption with these GMAs and PY-IT acceptor. Photovoltaic performance of the OSCs were optimized by using 1% 1-chloronaphthalene (1-CN) as solvent additive in chloroform solutions[25,31]. The detailed device fabrication procedures were described in Supplementary information. Figure 3a shows the current density-voltage ($J$-$V$) curves of the corresponding optimized devices, and the photovoltaic parameters were summarized in Table 2. $V_{oc}$ of the devices based on YDT, DY, TY, QY to PY-IT are 0.986 V, 0.959 V, 0.953 V, 0.937 V and 0.927 V, respectively. The slightly decreased $V_{oc}$ could result from the downshifted $E_{LUMO}$ of the acceptors and the varied $\Delta E_{loss}$ of the corresponding devices. The $\Delta E_{loss}$ of YDT, DY, TY, QY and PY-IT-based devices will be discussed later in the following. The device based on the GMA of TY shows the champion PCE of 16.32% with the best FF of 73.36% and higher $J_{sc}$ of 23.35 mA/cm² compared to the other devices. In addition, although the PY-IT-based device exhibits similar $J_{sc}$ to that of the TY-based OSC, its lower FF and $V_{oc}$ lead to the lower PCE. The poorest performance of the YDT-based device may be due mainly to the mismatched energy levels and the small excitons diffusion length mentioned above, which seriously affects the exciton dissociation and charge transport[40]. Figure 3b further exhibits the variation trend of the photovoltaic parameters of these OSCs based on the acceptors for a clear comparison. With the increased SMA subunits from YDT to PY-IT, the corresponding devices show gradually decreased $V_{oc}$, first raised and then descended FF, as well as the $J_{sc}$ increases first and then changes slightly. The combination of these parameters leads to the first increased and then decreased PCE values from YDT to PY-IT-based devices.

To further investigate the internal mechanism of the varied $V_{oc}$ values for these devices, the $\Delta E_{loss}$ measurements were conducted for the corresponding devices. The detailed description of the experiments are shown in the methods part. According to the energy loss measurement, the $\Delta E_1$ values are 0.265-0.267 eV for these five acceptors-based devices. The $\Delta E_2$ values are 0.078 eV for the YDT-based device, 0.062 eV for the DY-based device, 0.058 eV for the TY-based device and 0.022 eV for the QY-based device. The $\Delta E_3$ is non-radiative recombination loss and contributes to the largest part of the total energy loss in these devices. The values of $EQE_{EL}$ for this system determine the $\Delta E_3$ which were summarized in Supplementary Table 1, and the corresponding curves were exhibited in Supplementary Fig. 34. For the acceptors with determined structure, the $EQE_{EL}$ of these acceptors-based devices gradually decreases with the increased molecular size, leading to the increased non-radiative recombination loss of 0.150, 0.190, 0.202 and 0.253 eV for YDT, DY, TY and QY-based devices, respectively. The enlarged $\Delta E_3$ in the corresponding devices may be due to the increased energetic disorder (Supplementary Fig. 35), which may originate from the increased molecular size and lack of halogen substitution in the terminal. Thus, the above energy loss components result in the gradually increased $\Delta E_{loss}$ from 0.494 eV for the YDT-based device, to 0.519 eV for the DY-based device, 0.526 eV for the TY-based device and to 0.540 eV for the QY-based device. Compared to the SMA and GMAs, PY-IT with uncertain structure and wide molecular weight distribution exhibits an unusual

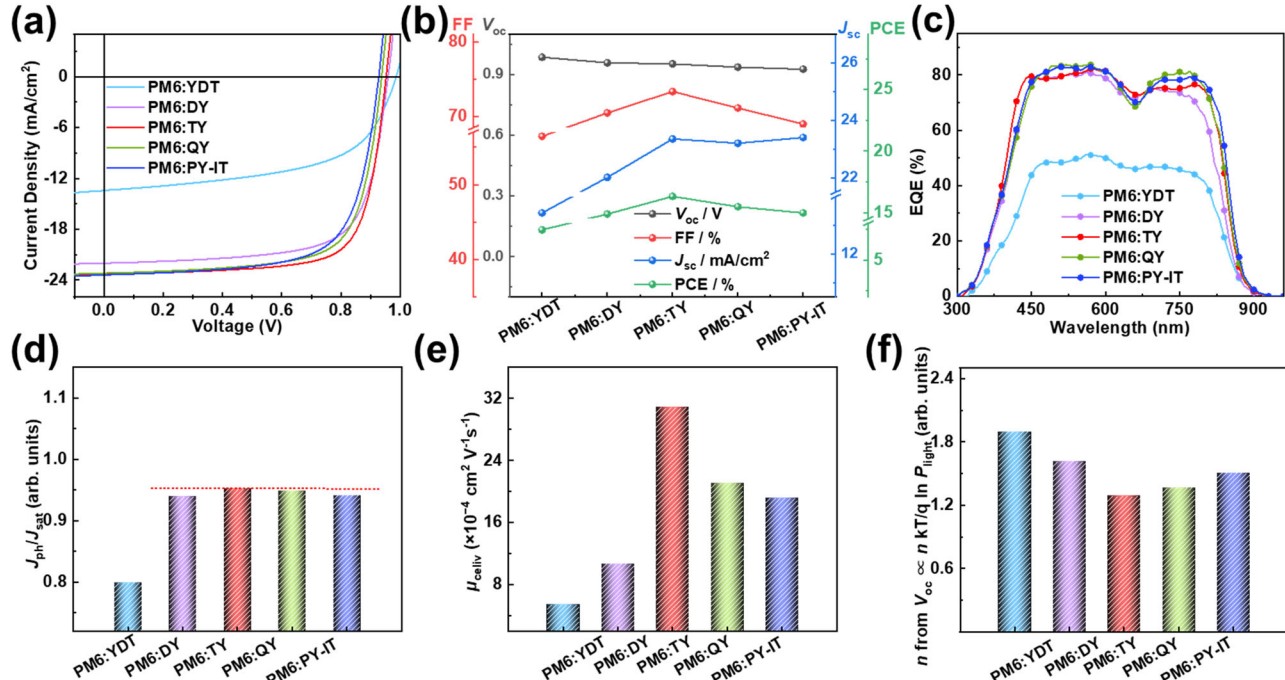

**Fig. 3 | Device performance of these acceptors. a** $J$-$V$ characteristics of the optimal OSCs based on PM6:acceptors under the illumination of AM1.5 G, 100 mW cm$^{-2}$. **b** The trend of the values for $V_{oc}$, FF, $J_{sc}$ and PCE various from YDT to PY-IT. **c** The optimal EQE curves of the corresponding OSCs based on PM6:acceptors. The comparisons between the five acceptors-based devices in **d** exciton dissociation and charge collection behaviors; **e** charge carrier transportation properties and **f** charge carrier recombination behaviors as obtained from $J_{ph}$ $vs$ $V_{eff}$ curves, Photo-CELIV curves and the dependences of $V_{oc}$ on light intensity, respectively. Source data are provided as a Source Data file.

**Table 2 | Photovoltaic parameters of the optimized OSCs based on PM6:acceptors under the illumination of AM 1.5 G (100 mW cm$^{-2}$)**

| Active layer | $V_{oc}$ (V) | FF (%) | $J_{sc}$ (mA cm$^{-2}$) | $J_{cal.}$[a)] (mA cm$^{-2}$) | PCE[b)] (%) |
|---|---|---|---|---|---|
| PM6:YDT | 0.986 (0.985 ± 0.002) | 56.44 (56.38 ± 0.25) | 13.43 (13.40 ± 0.06) | 13.30 | 7.47 (7.44 ± 0.05) |
| PM6:DY | 0.959 (0.957 ± 0.002) | 70.50 (70.27 ± 0.15) | 22.01 (21.94 ± 0.11) | 21.37 | 14.88 (14.75 ± 0.12) |
| PM6:TY | 0.953 (0.954 ± 0.002) | 73.36 (73.30 ± 0.17) | 23.35 (23.09 ± 0.18) | 22.52 | 16.32 (16.15 ± 0.21) |
| PM6:QY | 0.937 (0.937 ± 0.002) | 71.15 (70.32 ± 0.48) | 23.20 (23.03 ± 0.20) | 22.74 | 15.47 (15.17 ± 0.31) |
| PM6:PY-IT | 0.927 (0.927 ± 0.002) | 69.02 (68.93 ± 0.66) | 23.40 (23.35 ± 0.15) | 22.94 | 14.97 (14.92 ± 0.19) |

[a]The values were obtained from the integration of EQE curves.
[b]The averaged device parameters from more than 10 devices.

change in energy loss of its device. This results in a similar $\Delta E_{loss}$ of 0.536 eV for PY-IT-based device compared to that of the QY-based device (0.540 eV) although PY-IT may possess larger molecular size, which is probably due to the different interface morphology characteristics between PSMA and GMA-based blends[41]. Therefore, the varied $\Delta E_{loss}$ of these acceptors-based devices and the progressively down-shifted $E_{LUMO}$ from YDT to DY, TY, QY and PY-IT cooperatively result in the slightly decreased $V_{oc}$ from the YDT-based device to the DY, TY, QY and PY-IT based devices.

Figure 3c shows the external quantum efficiency (EQE) curves of the OSCs based on these acceptors. The OSCs based on SMA YDT, GMAs DY, TY and QY, and PSMA PY-IT exhibit a similar photo-response range from 450 to 900 nm. The integrated current densities ($J_{cal}$) of the devices based on the corresponding acceptors are 13.30 mA/cm$^2$ for the SMA YDT, 21.37, 22.52 and 22.74 mA/cm$^2$ for the GMAs of DY, TY and QY respectively, 22.94 mA/cm$^2$ for the PSMA PY-IT. These $J_{cal}$ values of the corresponding devices matched well with their $J_{sc}$ values obtained from the $J$-$V$ curves, indicating the reliability of the photovoltaic performance measurements. The difference in the $J_{cal}$ is mainly reflected in the photo-response of the acceptor part (from 670 to 870 nm) in the EQE curves, which indicates that the number of SMA

subunits has a significant effect on the photo-response properties of the corresponding OSCs.

**Exciton dissociation, charge transport and recombination properties**

The differences in photovoltaic performances of the OSCs based on these acceptors were further investigated by exploring their internal exciton dissociation, charge transport and recombination characteristics. The exciton dissociation of these blend films was investigated by their photoluminescence (PL) quenching efficiency. As shown in Supplementary Fig. 36, when excited at a wavelength of 700 nm, the PL peaks of the corresponding blend films with PM6 as the donor are quenched by 82% for YDT, 90% for DY, 94% for TY, 92% for QY and 94% for PY-IT, respectively, suggesting more effective hole transfer from the GMAs and PY-IT to PM6 than that from YDT to PM6. In addition, the curves of photocurrent density ($J_{ph}$) versus effective voltage ($V_{eff}$) for these OSCs (Supplementary Fig. 37) were investigated to further gain insights into the corresponding exciton dissociation and charge collection behavior[42,43]. The exciton dissociation and collection probability can be estimated from $J_{ph}/J_{sat}$. The calculated $J_{ph}/J_{sat}$ value is 0.953 for the TY-based device, which is the highest value among the

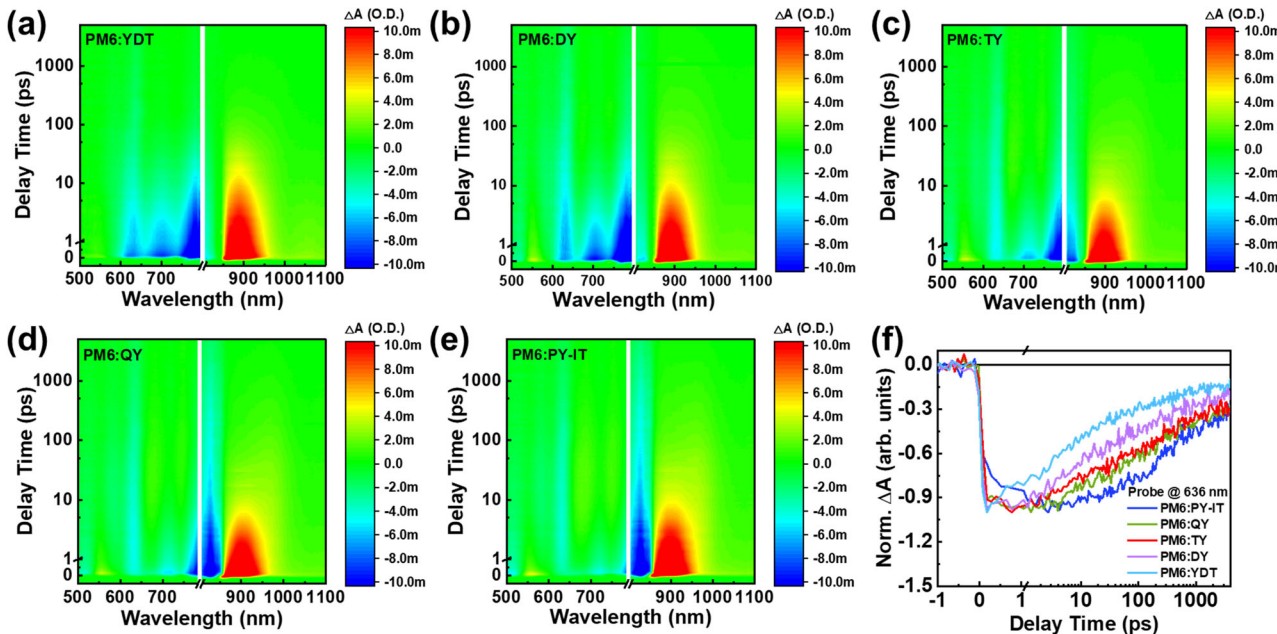

**Fig. 4 | Femtosecond transient absorption kinetics.** 2D transient absorption spectra of **a** PM6:YDT, **b** PM6:DY, **c** PM6:TY, **d** PM6:QY and **e** PM6:PY-IT. **f** Kinetic traces of PM6 GSB probing at 636 nm for the PM6:acceptor blend films. Source data are provided as a Source Data file.

three GMAs-based devices (0.940 and 0.949 for the DY and QY-based devices). Meanwhile, this also surpasses that value of 0.942 for the PY-IT-based device and 0.800 for the YDT-based device (see Fig. 3d). These results indicate the exciton dissociation and charge collection probability of the TY-based device outperforms those of the devices based on the other two GMAs (DY and QY) and the PSMA PY-IT and the SMA subunit YDT.

Furthermore, the charge transport properties of these acceptors were explored by the space charge limited current (SCLC) method for both neat and blend films. The corresponding fitting curves were exhibited in Supplementary Fig. 38a–c, and the calculated data are summarized in Supplementary Table 2. The electron mobilities of the acceptors neat films are $8.6 \times 10^{-4}$, $13.9 \times 10^{-4}$, $17.1 \times 10^{-4}$, $15.8 \times 10^{-4}$ and $14.4 \times 10^{-4}$ cm$^2$ V$^{-1}$s$^{-1}$ for YDT, DY, TY, QY and PY-IT, respectively. Among these acceptors, TY with three SMA subunits exhibits the best electron mobility. This may be mainly attributed to the fact that the increased molecular conjugation length will be beneficial to the charge transport, but too large molecular size may hinder the orderly packing of molecules in the film, thereby affecting the charge transport ability of the materials. Therefore, under the comprehensive factors, the electron mobility of these acceptors with increased molecular size shows a trend of first increasing, reaching a top value for TY and then decreasing. Subsequently, the charge carrier generation and transport behaviors of these acceptors-based devices under working conditions were further investigated via the photoinduced charge extraction by linearly increasing voltage (photo-CELIV)[44]. Supplementary Fig. 39a exhibits the photo-CELIV curves of these acceptors-based devices, from which the extracted charge carrier mobility $\mu_{\mathrm{celiv}}$ can be calculated and the corresponding values are $5.50 \times 10^{-4}$, $10.74 \times 10^{-4}$, $30.87 \times 10^{-4}$, $21.09 \times 10^{-4}$, and $19.21 \times 10^{-4}$ cm$^2$ V$^{-1}$s$^{-1}$ for the YDT-, DY-, TY-, QY- and PY-IT-based devices respectively (Fig. 3e), which shows the same trend as that measured by SCLC method. Overall, the TY-based device showed the higher charge mobilities, followed by that of the QY and DY-based devices, which all outperform those of the PY-IT and YDT-based devices. The trend of the charge mobilities of these acceptors shows a high degree of similarity with the variation of FF for the devices based on these acceptors, therefore, the charge mobility may be one of the main factors affecting the FF of the corresponding OSCs.

The charge carrier recombination behaviors of the OSCs were investigated by exploring the dependence of $J_{\mathrm{sc}}$ and $V_{\mathrm{oc}}$ on the illumination light intensity ($P_{\mathrm{light}}$)[45,46]. Supplementary Fig. 39b show the plots of $\log J_{\mathrm{sc}}$ versus $\log P_{\mathrm{light}}$ for these acceptors-based devices. Combining with the typical relationship of $J_{\mathrm{sc}} \propto P_{\mathrm{light}}^{\alpha}$, the values of $\alpha$ can be obtained from the slope of $\log J_{\mathrm{sc}}$ *vs* $\log P_{\mathrm{light}}$. Meanwhile, the value of $\alpha$ closer to 1 means the less bimolecular recombination. The TY-based device exhibits the highest $\alpha$ value of 0.989, and then the $\alpha$ value of the QY-based device is 0.984, both are higher than that of 0.977 for the DY-based device. Besides, the $\alpha$ values of the PY-IT and YDT-based devices are 0.982 and 0.956, which is also lower than those of the TY and QY-based devices. Furthermore, the dependence of $V_{\mathrm{oc}}$ on $\ln P_{\mathrm{light}}$ of these devices were also investigated, and the results are shown in Supplementary Fig. 39c. The slope of $V_{\mathrm{oc}}$ *vs.* $\ln P_{\mathrm{light}}$ is 1.90, 1.62, 1.29, 1.37 and 1.51 kT/q for the YDT, DY, TY, QY and PY-IT-based devices, respectively (Fig. 3f). Notably, the closer to 1 kT/q the slope is, the less trap-assisted charge recombination in the corresponding devices[47]. Thus, the TY-based OSC with the smaller slope of 1.29 kT/q indicates less trap-assisted charge recombination, while the YDT-based device with the large slope of 1.90 kT/q possessed serious trap-assisted charge recombination.

To better understand the effects of the number of SMA units of the acceptors on the charge transfer process, femtosecond transient absorption spectroscopy (fsTA) measurement was carried out on the PM6:acceptor blend films. Figure 4a and Supplementary Fig. 40a display fsTA spectra of PM6:YDT blend film at selected time delays. Pump wavelength was set to 820 nm to selectively excite the acceptor in the blend. Immediately after excitation, a strong excited state absorption (ESA) showed up at 900 nm and a series of negative ground state bleach (GSB) peaks were observed at 820 nm, 710 nm, resulting from the generation of YDT exciton. Another GSB that matches the PM6 absorption was simultaneously observed at 636 nm, which is due to the ultrafast hole transfer from photoexcited YDT to PM6, generating charge transfer (CT) state. Transient absorption spectra was also collected for the PM6:GMAs and PM6:PY-IT blend films and similar spectra profiles were also obtained except for the gradually red-shifted GSB peak of the acceptor around 820 nm (Fig. 4b–e and Supplementary Fig. 40b–e). Then we monitored the kinetic traces of the GSB of PM6 at

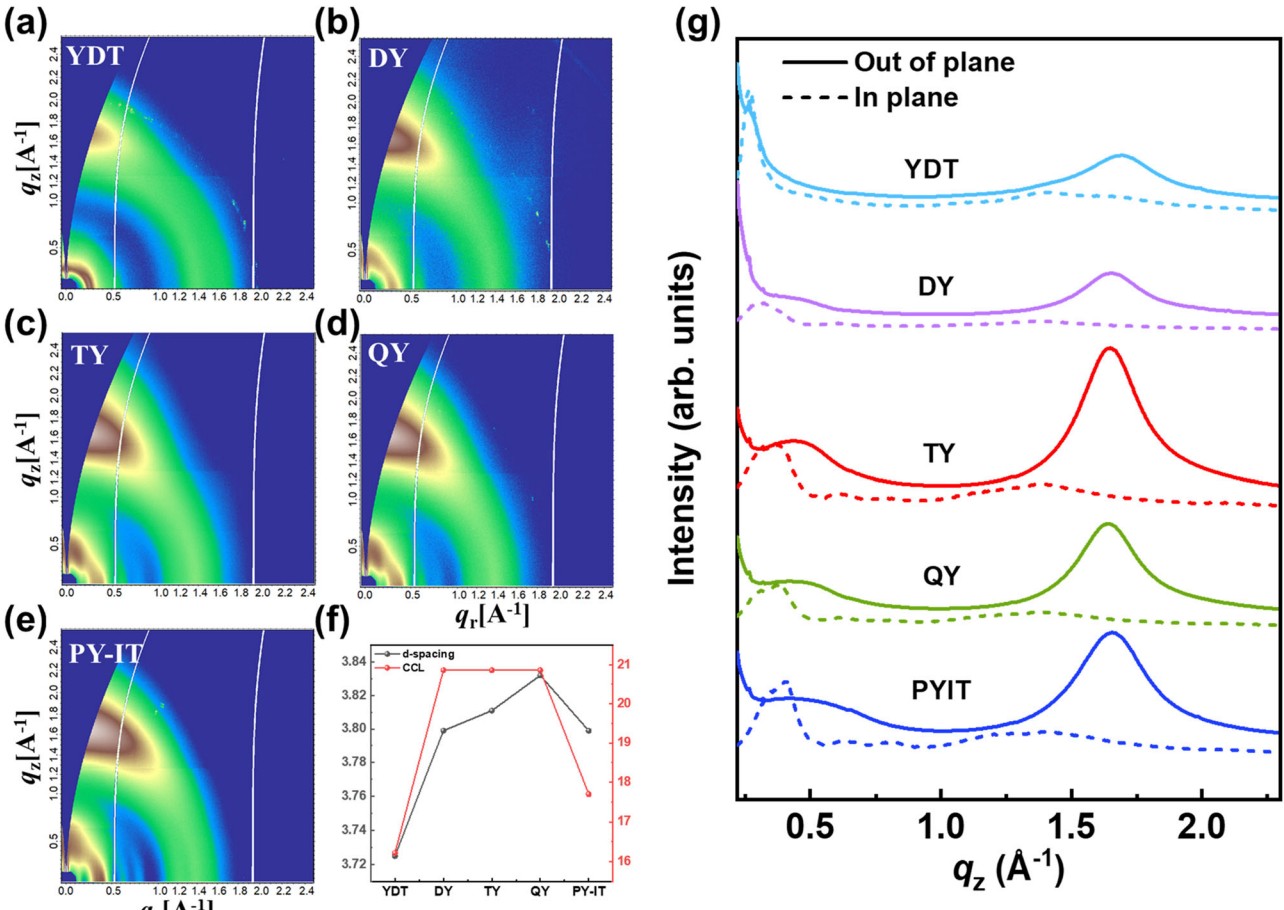

**Fig. 5 | Results of GIWAXS measurements.** 2D GIWAXS patterns of **a** YDT neat film, **b** DY neat film, **c** TY neat film, **d** QY neat film and **e** PY-IT neat film. **f** Molecular packing parameters of the neat films from GIWAXS fitting. **g** The corresponding in-plane and out-of-plane line-cut curves. Source data are provided as a Source Data file.

637 nm for all the blends to extract their hole transfer dynamics. As shown in Fig. 4f, the PM6 GSB of the PM6:YDT blend film directly decay to ground state after the initial CT, indicating a strong charge recombination in the active layers, which leads to lower $J_{sc}$ and FF in the devices. While the DY, TY, QY and PY-IT-based blend films exhibited a two-step CT process as the PM6 GSBs continued to grow and reach a maximum after the initial rise, indicating a diffusion-mediated CT process coming after the ultrafast CT at the D/A interface. The CT process tends to be slower from DY to PY-IT. Another trend is easily noticed that the charge recombination also becomes slower with the increase of the SMA units from YDT to PY-IT according to the decay rate of this signal. These findings suggest that CT properties are affected by the size of the acceptor molecules. YDT blend lacks diffusion-mediated CT and displays rapid charge recombination which might be ascribed to the over-mixing of the donor and acceptor and undersized D/A phase domains. On the contrary, the slow CT and charge recombination in PM6:PY-IT could be attributed to the large D/A phase domains, which is also observed in PiFM experiment as discussed in the next morphology characterization section. The PM6:TY and PM6:QY blend films reached a balance between relatively fast CT and slow charge recombination due to proper phase sizes. Therefore, the PM6:TY and PM6:QY blend films obtained a higher CT state yield and longer CT state lifetime, which agreed well with their better device performances.

## Morphology Characterization

Grazing-incidence wide-angle X-ray scattering (GIWAXS) measurements were conducted to explore the molecular stacking and morphology features of the neat and blend films of these acceptors. Figure 5a–e and g show the 2D GIWAXS patterns and the corresponding line-cut curves for the neat acceptor films, from which a dominant face-on orientation is inferred in these films. The locations of the π-π stacking (010) scattering peaks in the out-of-plane (OOP) direction were gradually shifted to lower $q$ values from YDT, DY, TY to QY, indicating that the increased molecular size may result in the increased π-π stacking distances. As a mixture with wide molecular weight distribution, the PSMA PY-IT shows the deviated π-π stacking behavior from those of GMAs and SMA with exact molecular structures. The crystalline coherence lengths (CCLs) of π-π stacking were obtained utilizing the Scherrer equation[48] and the detailed information was summarized in Supplementary Table 3. The three GMAs exhibit the similar CCLs of 20.9 Å, higher than those of 17.7 Å for PY-IT and 16.2 Å for YDT. These results indicate that an increasing number of SMA subunits will first promote the orderly packing of molecules with increased CCL of π-π stacking. However, with further increase in molecular size, a too large structure may hinder molecular accumulation, thereby reducing the molecular order. It is worth mentioning that TY film shows highest normalized integrated intensity of the π-π stacking (Fig. 5g). Combined with the result of CCL, TY neat film exhibits the most ordered molecular packing among the GMAs neat films and its corresponding PSMA PY-IT and SMA subunit YDT neat films. This can explain the highest electron mobility in the neat TY film measured by SCLC method. Meanwhile, the 2D GIWAXS patterns and 1D line-cut curves of the corresponding blend films are depicted in Supplementary Fig. 41a–f. The five blend films present predominant

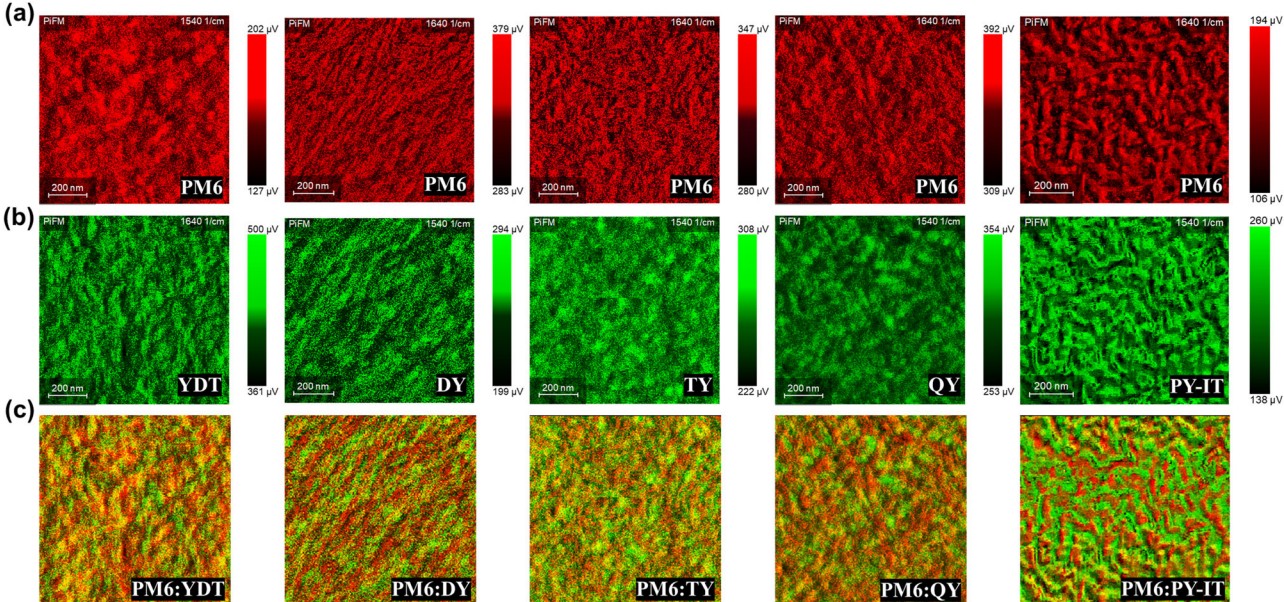

**Fig. 6 | PiFM images of the blend films.** The PiFM images of the PM6:acceptors blend films at the characteristic IR wavenumbers of 1640 cm$^{-1}$ for the **a** donor PM6 and of 1540 cm$^{-1}$ for the **b** acceptors YDT, DY, TY, QY and PY-IT. **c** the combined images of the corresponding **a** and **b** for PM6:YDT, PM6:DY, PM6:TY, PM6:QY and PM6:PY-IT blend films.

face-on orientation and their trend of π-π stacking distances is in line with those analyzed in neat films. Specifically, the π-π stacking distance is increased with the enlarged molecular size from YDT to DY, TY and QY, except for that of the polymer PY-IT. The integrated intensities of the π-π stacking of these blend films can be observed in the 1D line-cut curves (Supplementary Fig. 41f), from which the intensity of TY-based blend film outperforms other acceptor-based blend films. These results suggest that the TY-based blend film possesses higher volume fraction of ordered molecular packing, which contributes to the improved charge transport and the better device performance of the TY-based OSCs.

To further investigate the detailed donor/acceptor phase segregation morphology features of the blend films from SMA to GMAs and to PSMA with the gradually increased SMA subunits, the photoinduced force microscopy (PiFM) measurement was carried out to distinguish the donor and acceptor at nanometer scale and provide a desired contrast for chemical imaging with high spatial resolution for these blend films[49,50]. Owing to the similar chemical structure of these acceptors, their Fourier transform infrared (FTIR) absorption spectra are also similar, and they share the same characteristic peak of the acceptor components. The measured Fourier transform infrared (FTIR) absorption spectra of the donor PM6 and the acceptor YDT neat films are exhibited in Supplementary Fig. 42. The IR absorption wavenumber of 1640 cm$^{-1}$ and 1540 cm$^{-1}$ were chosen as the characteristic peaks of PM6 and YDT components respectively to distinguish the donor and acceptor phases in the blend films. The PiFM images of the donor and acceptor materials in the blend films were obtained from the atomic force microscopy (AFM) technology with the characteristic IR absorption signals acquired from the optically driven dipoles on the film surface. Figure 6a, b show the PiFM images of the donor (red color) and acceptor (green color) components measured from the YDT, DY, TY, QY and PY-IT-based blend films, respectively. The combined images of the donor and acceptor for the above-mentioned blend films are displayed in Fig. 6c. As the number of SMA subunits increases from YDT to DY, TY, QY and to PY-IT, the phase region of the acceptor in these blend films gradually expands (Fig. 6b), accompanied with a synchronously increased phase region of the donor (Fig. 6a). These correspond to the good miscibility between

PM6 and YDT with a relatively undersized phase domain, and the large phase separation between PM6 and PY-IT is estimated to own a large domain size of 30–50 nm. The D/A phase domains of the three GMAs DY, TY and QY-based blend films are in between those of YDT and PY-IT-based blend films. Notably, both too-small and too-large domains in the corresponding blend films are harmful to balance their charge transfer and recombination. Specifically, the undersized D/A phase domain in the YDT-based blend film leads to its rapid charge recombination, while the oversized D/A phase domain in the PY-IT-based blend film leads to its slow CT. The proper D/A phase separation in TY and QY-based blend films enables their relative balance between fast CT and slow charge recombination, contributing to a higher CT state yield and longer CT state lifetime, which will help to improve the photovoltaic performance of the TY- and QY-based OSCs.

## Stability measurement

The combination of high efficiency and excellent stability is the key to the commercialization of OSCs. The photostability of these acceptors were investigated roughly through UV-vis absorption measurement for the samples with one-sun-equivalent illumination in N$_2$ filled glove box. As shown in Supplementary Fig. 43, after 360 h illumination, all these acceptors exhibit preferable photostability without obvious absorption fluctuation.

Among the various intrinsic and extrinsic factors to influence the stability of OSCs, the metastable blend morphology of the active layer can strongly influence the stability of OSCs as the elevated temperature under illumination provides thermal energy and accelerates the diffusion of the acceptors. Herein, the diffusion capacity of these acceptors from YDT, to DY, to TY, to QY and to PY-IT needs to be investigated to better understand the stability of the corresponding acceptors-based OSCs. The $T_g$ of these materials was explored by the UV-vis absorption spectra of these acceptors in film at different annealing temperature (Supplementary Fig. 44)[51], which progressively increases with the gradually enlarged molecular size from 82 °C for YDT, to 132 °C for DY, to 190 °C for TY, to 210 °C for QY and to 239 °C for PY-IT. According to the Arrhenius diffusion behaviour, the diffusion coefficients ($D_{85}$) of these acceptors at 85 °C can be estimated based on their $T_g$ values[52]. The calculated $D_{85}$ values are 8.8 × 10$^{-17}$ cm$^2$ s$^{-1}$ for

PM6:YDT, $4.8 \times 10^{-20}$ cm$^2$ s$^{-1}$ for PM6:D, $8.1 \times 10^{-24}$ cm$^2$ s$^{-1}$ for PM6:TY, $4.0 \times 10^{-25}$ cm$^2$ s$^{-1}$ for PM6:QY and $5.2 \times 10^{-27}$ cm$^2$ s$^{-1}$ for PM6:PY-IT, respectively. Therefore, the larger molecular size enables the less molecular diffusion of acceptors into the donor polymer, which will help to improve the thermal stability of the corresponding active layers.

Furthermore, the thermal stability of these acceptors-based active layers was investigated by continuous annealing at 90 °C for different times, as shown in Supplementary Fig. 45. The suppressed diffusion of TY, QY and PY-IT contributes to the excellent thermal stability of their active layers for maintaining over 96% of their initial PCEs after continuous thermal annealing of active layers for 12 h, while the easier diffusion of DY and YDT leads to more fast decay of PCEs of the corresponding devices to drop below 95% and 90% respectively under the same condition. Considering the balance between the PCE and the stability of these devices, the GMA TY was chosen as the representor among the three GMAs DY, TY and QY to compare with the SMA YDT and PSMA PY-IT in their long-term devices' storage stability (Supplementary Fig. 46). This trend of storage stability of the OSCs based on YDT, TY and PY-IT is in line well with that of their thermal stability, which means that the larger molecular size of acceptor molecule helps to improve its morphology stability in the OSCs.

## Discussion

On the basis of SMA subunit of the representative PSMA PY-IT, a series of GMAs DY, TY and QY were synthesized by retrosynthetic method, with the boron trifluoride etherate-catalyzed Knoevenagel condensation and the classic Stille cross-coupling reaction. The SMA subunit YDT and the PSMA PY-IT were also synthesized for comparison. From SMA to GMA and then to PSMA, the main difference in structure is the gradually increased number of SMA subunits, accompanied with the more extended conjugated system, larger electron delocalization structure and gradually increased intermolecular distance. All these features synergistically contribute to their varied photoelectric properties and photovoltaic performances. It was found that the gradually increased molecular size and the extended conjugation from YDT to PY-IT leads to the gradually increased thermal stability, redshifted UV-vis absorption spectra and prolonged exciton diffusion length, which benefit to the improvement of the photovoltaic performance of the acceptors. While the $E_{LUMO}$ is down-shifted gradually from YDT to DY, TY, QY and PY-IT, accompanied with varied $\Delta E_{loss}$ of the devices, which results in the slightly decreased $V_{oc}$ of the corresponding devices. The proper D/A phase separation in TY and QY-based blend films also help to balance the CT and the charge recombination, which allows the TY- and QY-based blend films to show a relatively high CT state yield and long CT state lifetime. Interestingly, the GMA TY with three SMA subunits possesses the highest electron mobility which is two times of that of the SMA YDT and more than 10% higher than that of the PSMA PY-IT. Finally, the OSC based on the GMA TY exhibits the best PCE of 16.32%, with the best FF of 73.36% and higher $J_{sc}$ of 23.35 mA/cm$^2$, in comparison with the PCEs of 7.47%, 14.88%, 15.47% and 14.97% for the OSCs based on YDT, DY, QY and PY-IT respectively. These results suggest that the number of SMA subunits in the GMAs has a significantly important impact on the photovoltaic performances of these acceptors. It is worth noting that the larger molecular size of acceptors enables their less molecular diffusion coefficients, which helps to promote the thermal stability of the corresponding devices. In addition, from YDT to DY, TY, QY and to PY-IT, with the same structural subunit, the comparison of their physicochemical and photovoltaic properties provides an important reference for in-depth understanding of the structure-performance relationship from SMAs to PSMAs. Furthermore, the stepwise precise synthesis employed in this work can promote the diversification of GMAs and facilitates their application in highly efficient and stable OSCs.

## Methods

### Materials and synthesis

Polymer PM6 and the dialdehyde precursor were purchased from Solarmer Materials Inc. Other chemical reagents and solvents were also purchased from Innochem, J&K, Alfa Aesar, and TCI Chemical Co and used without further purification. The toluene agents for polymerizing were distilled from sodium and benzophenone under nitrogen before using. All reactions were performed under a nitrogen atmosphere.

The synthetic routes of these acceptors YDT, DY, TY, QY and PY-IT are shown in Fig. 1. The detailed synthetic procedures and characterizations of the chemical structures of the monomers and the polymer acceptors are described in the Supplementary method section. The NMR spectra and the MALDI-TOF-MS results of these compounds are shown in Supplementary Figs. 5–20 and 21–28.

### Molecular structure and property characterization

$^1$H and $^{13}$C NMR spectra were recorded on Avance 400 MHz NMR spectrometer and NEO 700 MHz NMR spectrometer. Chemical shifts are reported in parts per million (ppm, δ). $^1$H NMR and $^{13}$C NMR spectra were referenced to tetramethylsilane (TMS) (0 ppm) for CDCl$_3$. Mass spectra were collected on a Shimadzu spectrometer. Thermogravimetric analysis is conducted under N$_2$ atmosphere with 10 °C/min heating from 50 °C to 500 °C by Shimadzu DTG 60. UV-Vis absorption spectra of our materials were recorded on the Hitachi U-3010 UV-vis spectrophotometer. Gel permeation chromatography (GPC) measurements was performed on Agilent PL-GPC 220 instrument with high-temperature chromatograph, using 1,2,4-trichlorobenzene as the eluent at 160 °C.

UV-vis absorption spectra were recorded on the Hitachi U-3010 UV-vis spectrophotometer. For solution absorption, these materials were dissolved in chloroform. For the film measurements, the corresponding chloroform solutions were spin-coated on quartz plates.

Electrochemical cyclic voltammetry measurements were carried out in a conventional three-electrode cell, using a glass carbon electrode as the working electrode, a platinum wire as the counter electrode, and an Ag/AgCl electrode as the reference electrode. The photovoltaic materials were uniformly coated on the working electrode to form a thin film and the electrodes were placed in tetrabutylammonium hexafluorophosphate (Bu$_4$NPF$_6$) (0.1 mol L$^{-1}$) acetonitrile solution for testing, using a CHI660 Voltammetry Workstation. The energy levels were calculated using the following formula:

$$E_{HOMO}/E_{LUMO} = -e(\phi_{ox}/\phi_{red} + 4.8 - \phi_{Fc/Fc+})(eV) \quad (4)$$

the redox potentials of $\phi_{Fc/Fc+}$ vs Ag/AgCl was measured to be 0.44 V in our measurement system.

### Time-resolved photoluminescence spectroscopy

The TRPL spectra were recorded using the combination of a spectrometer (Princeton 200is) and a streak camera (C10910, Hamamatsu Photonics). Excitation light was generated using a fundamental pulse from Ti:sappaire regenerative amplifier (Astrella, Coherent) and then send to an optical parametric amplifier (OPerA-Solo, Coherent) for the pump beam with 700 nm wavelength. The software provided by Hamamatsu (HPD-TA, Hamamatsu Photonics) was used to collect and process the PL spectra.

### Determination of exciton diffusion length using singlet-singlet annihilation method

The PL decay with SSA is described by Eq. (1) with instantaneous generation rate:

$$\frac{dn(t)}{dt} = -kn(t) - \frac{1}{2}\gamma n^2(t) \quad (1)$$

For time-independent annihilation, the solution to Eq. (1) is:

$$n(t) = \frac{n(0)\exp(-kt)}{1 + \frac{\gamma}{2k}[1 - \exp(-kt)]} \qquad (5)$$

Where $n(t)$ is the singlet exciton density as a function of time after excitation, $k$ is the monomolecular decay rate constant which is the reciprocal of PL lifetime $\tau$, extracted from dilute acceptor samples, $n(0)$ is the initial exciton density, derived from the excitation fluence, extinction coefficient and laser spot size. Fitting the PL decays of the acceptor films at different excitation fluence using Eq. 5 could obtain the time-independent bimolecular annihilation rate constant $\gamma$. This constant is then used to determine the exciton diffusion coefficient $D$ and exciton diffusion length $L_D$.

## Fabrication and characterization of OSCs

The structure of all OSCs adopt the conventional device structure, namely ITO/PEDOT:PSS/active layer/PDINN/Ag structure. The pre-patterned ITO glasses substrate (sheet resistance = 15 $\Omega$ sq$^{-1}$) are sonicated sequentially with deionized water twice (with detergent and without), acetone and isopropanol in an ultrasonic bath. Before use, these glasses are dried in a vacuum oven and treated by UV-ozone (Jelight Company, USA) for 30 mins to improve its work function and clearance. Immediately, the PEDOT:PSS aqueous solution (Baytron P 4083, from HCStarck) is filtered by a 0.45 mm filter and pre-coated onto these pre-cleaned ITO glasses at 5000 rpm for 30 seconds. Then heat the ITO, then dried at 150 °C for 20 mins in air. The PEDOT: PSS coated ITO substrates were transferred to a N$_2$-filled glove box for further processing. The device was optimized according to the previous reported study with D:A weight ratio of 1:1, solution concentration of 14 mg/ml in chloroform with 1% 1-chloronaphthalene as additive. Then the solution was stirred for 2 hours for intensive mixing. The blend solutions were spin-coated on the PEDOT:PSS layer at 3000 rpm for 30 seconds, then annealed at 90 °C for 10 minutes. After cooling to room temperature, the PDINN methanol solution with a concentration of 1.0 mg mL$^{-1}$ was deposited on these active layers at 5000 rpm for 30 seconds. Then, a Ag layer (-100 nm) was deposited in thermal evaporator under vacuum of $5 \times 10^{-5}$ Pa through a shadow mask. The active area of the OSCs was 6.0 mm$^2$ (3 mm × 2 mm), which was defined by Optical microscope (Olympus BX51). In order to accurately measure the photocurrent, mask with an area of 4.16 mm$^2$ (0.26 mm × 0.16 mm) was used to define the effective area of the OSCs. The devices with or without mask showed consistent photovoltaic performance values with relative errors within 0.3%. The current density–voltage ($J$-$V$) characteristics of OSC are measured in a N$_2$-filled glove box equipped with a Keithley 2450 Source Measure Unit, using Oriel Sol3A Class AAA Solar Simulator (model, Newport 94023 A) with 450 W xenon lamp and air quality (AM) 1.5 filter as the light source. The light intensity is calibrated to 100 mW cm$^{-2}$ by Newport Oriel 91150 V reference cell. The external quantum efficiency (EQE) value is measured by the solar cell spectral response measurement system QE-R3-011 (Taiwan Enli Technology Co., Ltd.). Standard single-crystal silicon photovoltaic cells are used to calibrate the light intensity of each wavelength.

## Energy loss measurements

Fourier-transform photocurrent spectroscopy external quantum efficiency (FTPS-EQE) was measured by using an integrated system (PECT600, Enlitech). External electroluminescence quantum efficiency (EQE$_{EL}$) measurements were performed by applying external voltage/current sources through the devices (REPS, Enlitech). The total $\Delta E_{loss}$ is determined by the optical gap $E_g$ and $V_{oc}$ ($\Delta E_{loss} = E_g - qV_{oc}$), which can be divided into three parts according to the Shockley-Queisser (SQ) limit[53]:

$$\Delta E_{loss} = \Delta E_1 + \Delta E_2 + \Delta E_3 \qquad (6)$$

The $\Delta E_1$ is the unavoidable radiative recombination losses above the bandgap and is decided only by the $E_g^{pv}$ of the absorber and temperature:

$$\Delta E_1 = E_g - qV_{oc}^{SQ} \qquad (7)$$

$V_{oc}^{SQ}$ is the maximum $V_{oc}$ in the SQ limit. The $\Delta E_2$ stands for the radiative recombination losses below the bandgap and is ascribed to non-step function like absorption or EQE of the real-world devices:

$$\Delta E_2 = qV_{oc}^{SQ} - qV_{oc}^{rad} \qquad (8)$$

$V_{oc}^{rad}$ is the $V_{oc}$ where only radiative recombination occurs. The $\Delta E_3$ is non-radiative recombination loss:

$$\Delta E_3 = -k_B T\ln(EQE_{EL}) \qquad (9)$$

EQE$_{EL}$ is the radiative quantum efficiency of the device when charge carriers are injected into the device in the dark.

## Charge property characterization

The dependence of the photocurrent density ($J_{ph}$) on the effective voltage ($V_{eff}$) was also tested to analyze exciton dissociation and charge collection. $J_{ph}$ was obtained from $J_{ph} = J_L - J_D$ ($J_L$ and $J_D$ are the current densities under illumination and dark conditions), and $V_{eff}$ was calculated from $V_{eff} = V_0 - V$ ($V_0$ is the voltage when $J_L = J_D$ and $V$ is the applied voltage). Under short-circuit conditions, the exciton dissociation and collection probability = $J_{ph}/J_{sat}$, where $J_{sat}$ represents the saturation photocurrent density.

The electron and hole mobility were measured by using the method of space-charge limited current (SCLC), ITO/PEDOT:PSS/active layer/MoO$_3$/Ag device structure is used to test hole mobility, and ITO/ZnO/active layer/PDINN/Al is used to test electron mobility. The hole and electron mobilities are calculated according to the space charge limited current (SCLC) method with the equation:

$$J = 9\mu\varepsilon_r\varepsilon_0 V^2/8d^3 \qquad (10)$$

where $J$ is the current density, $\mu$ is the hole or electron mobility, $V$ is the internal voltage in the device, $\varepsilon_r$ is the relative dielectric constant of active layer material, $\varepsilon_O$ is the permittivity of empty space, and $d$ is the thickness of the active layer.

The $J_{sc}$ and $V_{oc}$ under different light intensity. In organic solar cell devices, the relationship between $J_{sc}$ and $P_{light}$ can be expressed by the formula $J_{sc} \propto (P_{light})^\alpha$, where $\alpha$ is the power exponent. When the value of $\alpha$ in the formula approaches 1, the bimolecular recombination in the device can be ignored. The $V_{oc}$ and $\ln(P_{light})$ can be fitted to a straight line, and the slope of the fitted line should be kT/q (where k is Boltzmann's constant, T is the Kelvin temperature, and q is the elementary charge)

Photogenerated charge extraction by linearly increasing voltage (photo-CELIV) mobilities data was obtained by the all-in-one characterization platform, Paios (Fluxim AG, Switzerland). In the photo-CELIV measurement, the delay time is set to 0 s, the light intensity is 100%, the light-pulse length is 100 μs, finally the sweep ramp rate rises from 20 V/ms to 100 V/ms. The devices are the same as stated above.

## Transient absorption spectroscopy

Femtosecond transient absorption spectrometer was composed of a regenerative-amplified Ti: sapphire laser system (Coherent) and Helios

pump-probe system (Ultrafast Systems). The regenerative-amplified Ti: sapphire laser system (Legend Elite-1K-HE, center wavelength of 800 nm, pulse duration of 25 fs, pulse energy of 4 mJ, repetition rate of 1 kHz) was seeded with a mode-locked Ti: sapphire laser system (Vitara) and pumped with a Nd: YLF laser (Evolution 30). The output 800 nm fundamental of the amplifier was split into two beam pulses. The main part of the fundamental beam went through the optical parametric amplifiers (TOPAS-C), whose output light was set as the pump light with wavelength of 820 nm and chopped by a mechanical chopper operating at frequency of 500 Hz. A small part of the fundamental beam was introduced into the TA spectrometer in order to generate the probe light. After passing through a motorized optical delay line, the fundamental beam was focused on a sapphire crystal or YAG crystal, which was used to generate the white-light continuum (WLC) probe pulses with wavelength of 430 to 820 nm or 800 to 1600 nm, respectively. The optical path difference between the pump light and the probe light, which is controlled by the motorized optical delay-line, was used to monitor the transient states at different pump-probe delay. A reference beam was split from the WLC in order to correct the pulse-to-pulse fluctuation of the WLC. The pump was spatially and temporally overlapped with the probe beam on the sample. Excitation energy of the pump pulse was set to $2\,\mu J/cm^2$ to avoid singlet-singlet annihilation. The film samples for TA measurements were prepared by spin-coating the corresponding materials on thin quartz plates. The film samples were thermally annealed the same way as the actual device.

## Morphological characterization

Grazing-incidence wide-angle X-ray scattering (GIWAXS) measurements were conducted at Advanced Light Source (ALS), Lawrence Berkeley National Laboratory, Berkeley, CA at the beamline 7.3.3[54]. Data was acquired at the critical angle (0.16°) of the film with a hard X-ray energy of 10 keV. X-ray irradiation time was 30–60 s, dependent on the saturation level of the detector. The scattered intensity was detected with a Pilatus detector. 1D profile was obtained with the intensity distribution analyzed along in-plane and out-of-plane direction. Crystal coherence lengths (CCL) are estimated based on the Scherrer equation ($L = 2\pi K/FWHM$), where K is the shape factor (here we use 0.9), and FWHM is the full width at half maximum of diffraction peaks.

## Photo-induced force microscope (PiFM)

The microscope used is a VistaScope from Molecular Vista, Inc., operated in dynamic mode using commercial gold-coated silicon cantilevers (NCHAu) from Nanosensors. The excitation laser is a Laser Tune IR Source from Block Engineering.

## Reporting summary

Further information on research design is available in the Nature Portfolio Reporting Summary linked to this article.

## Data availability

The data that support the findings of this study are presented in Supplementary Information and Source Data file. The source data for Figs. 2–5, Supplementary Figs. 37, 39 generated in this study are provided with this paper. Source data are provided with this paper.

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

## Acknowledgements

This work was supported by the National Key Research and Development Program of China (No. 2022YFF0709903 (X.L.), 2019YFA0705900 (L.M.)) funded by MOST, NSFC (Nos. Nos. 51820105003 (Y.L.), 21734008 (Y.L.), 52203248 (X.L.), 61904181 (L.M.), 52103243 (J.Z.) and 52173188 (L.M.)) the Key Research Program of the Chinese Academy of Sciences (No. XDPB13) (J.H.) and the Basic and Applied Basic Research Major Program of Guangdong Province (No. 2019B030302007 (Y.L.)). GIWAXS data were acquired at beamline 7.3.3 at the Advanced Light Source, which is supported by the Director, Office of Science, Office of Basic Energy Sciences, of the U.S. Department of Energy under Contract No. DE-AC02-05CH11231 (D.A.). Work by NCSU supported by ONR grant N000142012155 (D.A.). The energy loss measurements were helped by Ruimin Zhou, work in College of Chemistry and Green Catalysis Center, Zhengzhou University, Zhengzhou 450001, China.

## Author contributions

H.Z. synthesized and characterized these acceptors Y.D.T., D.Y., T.Y., Q.Y. and P.Y.-I.T., and optimized the devices of these acceptors based OSCs. X. L. designed and directed all the experiment. J.Z. and J.L. conducted the TA measurements and data analysis. J. Z. carried out the TRPL measurements and analyzed the data. C.Z. conducted the PiFM measurements. H.H. participated in the synthesis of these acceptors. K.D. and H.A. measured and analyzed the GIWAXS. All authors contributed to data analysis, discussed the results, and commented on the manuscript. Y.L., X.L. and L.M., supervised the project, and H.Z., X.L., J.Z., and Y.L., wrote the paper.

## Competing interests

The authors declare no competing interests.
