## [Peer Review File · Nature Communications]

Precise synthesis and photovoltaic properties of giant molecule acceptorsREVIEWER COMMENTS

Reviewer #1 (Remarks to the Author):

This manuscript demonstrates the structure-performance relationship of bulk heterojunction organic photovoltaics to the molecular weight of non-fullerene acceptors by varying the number of small molecule acceptor subunits. The author reports that giant molecule acceptors face challenges of functionalizing more than two small molecule subunits, which the problem was tackled by retrosynthetic analysis to select suitable synthetic routes. The synthesis is achieved using boron trifluoride etherate-catalyzed Knoevenagel condensation and Stille cross-coupling reaction. Overall, the performance is only comparable to the current state-of-the-art, Y6, but this manuscript showcases the importance of balancing the molecular size (between small molecule to polymer) to fine-tune the size of the D/A phase domains to have an optimized rate of charge transfer and charge recombination. The gradual increase of molecular weight had minimal effect on their energetics, which supports further that the device performance is heavily impacted by their sizes and degree of mixing with a polymer donor. The materials are well characterized, and the blend films for small molecules to giant molecules to polymer with donor polymers were also well characterized using various techniques clearly showing the degree of phase mixing and charge transfer/recombination. I find no technical flaws with the manuscript, and it is suitable for publication.

An issue I find is conceptually. How are these new “larger molecules” better than Y6 non-fullerene acceptor. Indeed, the monomer YDT only gives 7% PCE devices, less than half of Y6 devices. Clearly the use of F-atoms on the endcaps is important. A direct comparison to Y6 should be made and advantageous that warrant the extra synthetic steps discussed.

Beyond comparing optoelectronic properties what advantages do these giant molecules bestow on the organic solar cell device. Does this increased molecular weight improve the strength of the BHJ blend and thus improve the bendability or stretchability of the devices? Does using higher MW non-fullerene acceptors lead to more viscous inks and better film formation using R2R methods? Chloroform was used as the solvent, what about green solvents? Is the long-term stability of the devices increased? I did see any stability data; this should be added.

For publication in Nature, I would expect a clear innovation and reason for using the new materials. The oligomer approach is well known and often studied, thus please present a case as to why these molecules are superior to Y6 which simply works so well as a non-fullerene acceptor in organic solar cell devices.

Reviewer #2 (Remarks to the Author):

Li et al. successfully synthesized a series of giant molecule acceptors with different numbers of small molecule acceptors through retrosynthetic method. Subsequently, the effect of the number of small molecule acceptors on the physicochemical and photovoltaic properties of giant molecule acceptors was systematically studied. It is revealed that the gradually extending conjugated structure shows the prolonged exciton diffusion length and red-shifted UV-vis absorption spectra. After blending with PM6, the TY-based OSC achieves best PCE of 16.32% due to proper D/A phase separation, higher CT state yield and longer CT state lifetime. While the PCE reported in this manuscript is not as high as that for similar molecules including that for giant molecule acceptors (18.8%, *Adv. Energy Mater.*, 2023, 2301283) and that based on PY-IT has also exceeded (19%, *Nat. Commun.*, 2023, 14, 4148), the synthetic approach and the related structure-property results would be very helpful for the design and synthesis of more efficient (giant) OPV molecules. Hence, I would like to recommend this paper to be published after the following issues addressed:

1. The authors claim that PY-IT based blend films have the largest phase separation size which is not conducive to exciton dissociation. However, the PY-IT-based OSC shows the largest J_{sc} , and the authors need to give a proper explanation for this.
2. The PL quenching efficiency should be measured to investigate the exciton dissociation efficiency.
3. The temperature-dependent UV-vis absorption spectra of the acceptors is suggested to be added to investigate the molecular aggregation of these acceptors.
4. Large energy loss is a key factor limiting the PCE of OPV. Therefore, the energy loss of devices based on these acceptors needs to be measured.
5. The photostability and thermal stability of these devices should be measured.

Reviewer #3 (Remarks to the Author):

In the submitted manuscript entitled "Precise synthesis and photovoltaic properties of giant molecule acceptors", a stepwise precise synthesis method was developed to synthesize giant molecular acceptors (GMAs) with three or four acceptor subunits. This work provides a feasible way to diversify the GMAs for

the following study. Furthermore, the impact of the structural transition from small molecules to GMAs on their structure-property relationship was carefully investigated. The development of GMAs may provide an alternative and promising approach to advance the practical application of organic solar cells. Overall, this work represents the research interests of the community and deserves to be published in Nature Communications after addressing the following comments.

1. These acceptor-based devices show a trend of decreased Voc from YDT-based devices to DY, TY, and QY-based devices and then to PY-IT-based devices. The author mentioned that “The slightly decreased Voc could result from the downshifted ELUMO of the corresponding acceptors.” Energy loss also plays an important role in affecting the Voc of the corresponding device. The authors should quantify the energy losses in the corresponding devices.

2. From solutions to films, the maximum absorption peaks of these molecules present different degrees of redshift, which was explained by the different molecular aggregation characteristics of these acceptors. The molecular preaggregation behaviors of these acceptors in solutions may also affect their absorption properties. The authors are suggested to investigate the molecular preaggregation features in different solvents.

3. In the fs-TA measurement, the charge recombination becomes slower with the increase of the SMA units from YDT to PY-IT, which is not matched with the charge carrier recombination behaviors of these OSCs, as investigated by exploring the dependence of Jsc and Voc on the illumination light intensity (Plight). Please provide more evidence or explanation.

4. From SMA to GMA and then to PSMA, the gradually increased number of SMA subunits in these acceptors contributes to their gradually enlarged electron delocalization structure, which is beneficial for the corresponding charge transport. The authors are suggested to provide comprehend discussion and explain why the TY-based device shows the highest electron mobility among all these acceptors-based devices.

5. The authors are suggested to discuss more about the exciton lifetime. For instance, the shorter exciton lifetime vs. the longer exciton diffusion length.

Oct. 4, 2023

We have revised the manuscript according to the reviewers' comments and revision opinions. The main revisions are highlighted by red words. The following is our Response to the reviewers' comments:

Response to Reviewer #1:

This manuscript demonstrates the structure-performance relationship of bulk heterojunction organic photovoltaics to the molecular weight of non-fullerene acceptors by varying the number of small molecule acceptor subunits. The author reports that giant molecule acceptors face challenges of functionalizing more than two small molecule subunits, which the problem was tackled by retrosynthetic analysis to select suitable synthetic routes. The synthesis is achieved using boron trifluoride etherate-catalyzed Knoevenagel condensation and Stille cross-coupling reaction. Overall, the performance is only comparable to the current state-of-the-art, Y6, but this manuscript showcases the importance of balancing the molecular size (between small molecule to polymer) to fine-tune the size of the D/A phase domains to have an optimized rate of charge transfer and charge recombination. The gradual increase of molecular weight had minimal effect on their energetics, which supports further that the device performance is heavily impacted by their sizes and degree of mixing with a polymer donor. The materials are well characterized, and the blend films for small molecules to giant molecules to polymer with donor polymers were also well characterized using various techniques clearly showing the degree of phase mixing and charge transfer/recombination. I find no technical flaws with the manuscript, and it is suitable for publication.

1. *An issue I find is conceptually. How are these new "larger molecules" better than Y6 non-fullerene acceptor. Indeed, the monomer YDT only gives 7% PCE devices, less than half of Y6 devices. Clearly the use of F-atoms on the endcaps is important. A direct*

comparison to Y6 should be made and advantageous that warrant the extra synthetic steps discussed.

Response: Thanks for the reviewer's suggestion. The F-atoms on the endcaps of the acceptors have a great impact on their energy levels and molecular stacking properties, the difference of energy levels and molecular stacking between YDT and Y6 leading to the poor PCE of the YDT based device than that of Y6. In this manuscript, to maintain the structure similarity and compare with the polymer PY-IT, the non-F-atoms terminalized molecules YDT, DY, TY and QY were chosen to be the small molecule acceptor (SMA) and the giant molecule acceptors (GMAs) respectively. In addition, there are direct comparison between the F-atoms terminalized SMAs and GMAs in other studies. (*Energy Environ. Sci.* **2023** <https://doi.org/10.1039/D3EE00272A>; *Angew. Chem. Int. Ed.* **2023** <https://doi.org/10.1002/anie.202308595>). Although the GMAs need more extra synthetic steps than the SMAs, the devices based on the GMAs TYT and Tri-Y6-OD show simultaneously improved photovoltaic performance and higher stability as comparing to those devices based on the SMAs MYT and Y6-OD. Therefore, it is meaningful to develop GMAs for high performance and stable OSCs.

2. Beyond comparing optoelectronic properties what advantages do these giant molecules bestow on the organic solar cell device. Does this increased molecular weight improve the strength of the BHJ blend and thus improve the bendability or stretchability of the devices? Does using higher MW non-fullerene acceptors lead to more viscous inks and better film formation using R2R methods? Chloroform was used as the solvent, what about green solvents? Is the long-term stability of the devices increased? I did see any stability data; this should be added.

Response: Thanks for the reviewer's suggestion.

(1). The GMAs have been reported to increase the formation of tie chains in the BHJ blend films, which help to enhance the mechanical robustness of BHJ blend films and achieve the high-performance flexible OSCs. (*Adv. Mater.* **2023** <https://doi.org/10.1002/adma.202305562>; *Angew. Chem. Int. Ed.* **2023**

<https://doi.org/10.1002/anie.202310034>).

(2). Owing the GMAs have more inhibited excessive molecular aggregation property than the SMAs during the non-halogenated solvent processing, our group reported the highly efficient non-halogenated solvent processed OSCs based on the GMAs (*Angew. Chem. Int. Ed.* **2023**, 135, 202303551).

(3). As for the long-term stability issue, the larger molecular size of acceptor also helps to limit its diffusion into the donor polymer, therefore increase the thermal stability of the devices. According to the reviewer's suggestion, we explored the T_g of YDT, DY, TY, QY and PY-IT and estimated the diffusion coefficients of these molecules. We reported the results in the revised manuscript on pp. 23-24: "The T_g of these materials was explored by the UV-vis absorption spectra of these acceptors in film at different annealing temperature (**Supplementary Fig. 44**)⁵¹, which progressively increases with the gradually enlarged molecular size from 82 °C for YDT, to 132 °C for DY, to 190 °C for TY, to 210 °C for QY and to 239 °C for PY-IT. According to the Arrhenius diffusion behavior, the diffusion coefficients (D_{85}) of these acceptors at 85 °C can be estimated based on their T_g values⁵². The calculated D_{85} values are $8.8 \times 10^{-17} \text{ cm}^2 \text{ s}^{-1}$ for PM6:YDT, $4.8 \times 10^{-20} \text{ cm}^2 \text{ s}^{-1}$ for PM6:DY, $8.1 \times 10^{-24} \text{ cm}^2 \text{ s}^{-1}$ for PM6:TY, $4.0 \times 10^{-25} \text{ cm}^2 \text{ s}^{-1}$ for PM6:QY and $5.2 \times 10^{-27} \text{ cm}^2 \text{ s}^{-1}$ for PM6:PY-IT, respectively. Therefore, the larger molecular size enables the less molecular diffusion of acceptors into the donor polymer, which will help to improve the thermal stability of the corresponding active layers." (The added references are: [51] Qin, Y., *et al.* The performance-stability conundrum of BTP-based organic solar cells. *Joule* **5**, 2129-2147 (2021). [52] Ghasemi, M., *et al.* A molecular interaction-diffusion framework for predicting organic solar cell stability. *Nat. Mater.* **20**, 525-532 (2021).)

Meanwhile the thermal stability of photovoltaic performance of the OSCs based on PM6:acceptors were investigated. And the results were reported on pp. 24: "Furthermore, the thermal stability of these acceptors-based active layers was investigated by continuous annealing at 90 °C for different times, as shown in **Supplementary Fig. 45**. The suppressed diffusion of TY, QY and PY-IT contributes to

the excellent thermal stability of their active layers for maintaining over 96% of their initial PCEs after continuous thermal annealing of active layers for 12h, while the easier diffusion of DY and YDT leads to more fast decay of PCEs of the corresponding devices to drop below 95% and 90% respectively under the same condition. Considering the balance between the PCE and the stability of these devices, the GMA TY was chosen as the representative sample among the three GMAs DY, TY and QY to compare with the SMA YDT and PSMA PY-IT in their long-term storage stability of the devices (**Supplementary Fig. 46**). This trend of storage stability of the OSCs based on YDT, TY and PY-IT is in line well with that of their thermal stability, which means that the larger molecular size of the GMA TY molecules helps to improve its morphology stability in the OSCs.”

In addition, we have added “**decreased diffusion coefficients,**” in the “Abstract” part on p. 2, “**decreased molecular diffusion**” and “**and device stability**” in the “Introduction” part on p. 5, line 3 and line 18 respectively, and “**It is worth noting that the larger molecular size of acceptors enables their less molecular diffusion coefficients, which helps to promote the thermal stability of the corresponding devices.**” on p.25.

Supplementary Fig. 44. Plots of DM_T of (a) YDT, (b) DY, (c) TY, (d) QY and (e) PY-IT films as a function of annealing temperature.

Supplementary Fig. 45. The thermal stability of photovoltaic performance of the OSCs based on PM6:acceptors active layers under the continuous thermal annealing at 90 °C.

Supplementary Fig. 46. The PCE decay curves of the OSCs based on PM6:YDT, PM6:TY and PM6:PY-IT under different aging time at the room temperature in nitrogen filled glove box.

3. *For publication in Nature, I would expect a clear innovation and reason for using the new materials. The oligomer approach is well known and often studied, thus please present a case as to why these molecules are superior to Y6 which simply works so well as a non-fullerene acceptor in organic solar cell devices.*

Response: Thanks for the reviewer's suggestion. The power conversion efficiencies of the OSCs based on Y6 have reached a high level, while the combination of high

efficiency and long-term stability is still a major obstacle in realizing the commercialization of the OSCs. The development of GMAs has been reported to be helpful in improving the long-term stability and maintain the high efficiency of OSCs. Besides, GMAs also help to enhance the mechanical robustness of BHJ blend films and achieve the high-performance flexible OSCs. The more inhibited excessive aggregation of GMAs than that of SMAs during the non-halogenated solvent processing contributes to the corresponding highly efficient non-halogenated solvent processed OSCs (see Ref[20] in this manuscript). Therefore, the studies on the GMAs are very important for the practical applications of OSCs.

Response to Reviewer #2:

Li et al. successfully synthesized a series of giant molecule acceptors with different numbers of small molecule acceptors through retrosynthetic method. Subsequently, the effect of the number of small molecule acceptors on the physicochemical and photovoltaic properties of giant molecule acceptors was systematically studied. It is revealed that the gradually extending conjugated structure shows the prolonged exciton diffusion length and red-shifted UV-vis absorption spectra. After blending with PM6, the TY-based OSC achieves best PCE of 16.32% due to proper D/A phase separation, higher CT state yield and longer CT state lifetime. While the PCE reported in this manuscript is not as high as that for similar molecules including that for giant molecule acceptors (18.8%, Adv. Energy Mater., 2023, 2301283) and that based on PY-IT has also exceeded (19%, Nat. Commun., 2023, 14, 4148), the synthetic approach and the related structure-property results would be very helpful for the design and synthesis of more efficient (giant) OPV molecules. Hence, I would like to recommend this paper to be published after the following issues addressed:

1. The authors claim that PY-IT based blend films have the largest phase separation size which is not conducive to exciton dissociation. However, the PY-IT-based OSC shows the largest J_{sc}, and the authors need to give a proper explanation for this.

Response: Thanks for the reviewer's suggestion. Firstly, owing to the progressively increased exciton diffusion length L_D of these acceptors from 12.5 ± 0.5 nm for YDT film to 19.1 ± 0.6 nm for DY film, to 27.8 ± 1.1 nm for TY film, to 32.9 ± 0.9 nm for QY film and to 34.5 ± 1.2 nm for PY-IT film, the allowable phase separation size of these blend films can be also increased from YDT to DY, to TY, to QY and to PY-IT based blend films, which may be at least twice larger than the corresponding L_D . The phase separation size of PY-IT based blend film measured by PiFM is estimated to be in the range of 30-50 nm. This is still within the allowable phase separation size and will not affect the exciton transport and dissociation in PY-IT based blend films. Secondly, as reflected by the photoluminescence (PL) quenching efficiency and the photocurrent density (J_{ph}) versus effective voltage (V_{eff}) of the DY, TY, QY and PY-IT based OSCs, the PY-IT-based device exhibits very efficient exciton dissociation. Therefore, the largest phase separation has no effect on the exciton dissociation of the PY-IT based device. Furthermore, the most red-shifted UV-vis absorption spectra and the largest film extinction coefficient of PY-IT based film may further enhance the light harvest of the corresponding film, contributing to the higher J_{sc} in the PY-IT based OSC.

2. *The PL quenching efficiency should be measured to investigate the exciton dissociation efficiency.*

Response: Thanks for the reviewer's suggestion. The PL quenching efficiency of these acceptors-based blend films were measured and the results were reported on p.15: “**The exciton dissociation of these blend films was investigated by their photoluminescence (PL) quenching efficiency. As shown in Supplementary Fig. 36, when excited at a wavelength of 700 nm, the PL peaks of the corresponding blend films with PM6 as the donor are quenched by 82%, 90%, 94%, 92% and 94% for the blend films based on YDT, DY, TY, QY and PY-IT, respectively, suggesting more effective hole transfer of the GMAs and PY-IT to PM6 than that of YDT to PM6.**”

Supplementary Fig. 36. Photoluminescence spectra of the neat and blend films (with PM6 polymer donor) of (a) YDT, (b) DY, (c) TY, (d) QY and (e) PY-IT excited at 700 nm.

3. The temperature-dependent UV-vis absorption spectra of the acceptors are suggested to be added to investigate the molecular aggregation of these acceptors.

Response: Thanks for the reviewer’s suggestion. The temperature-dependent UV-vis absorption spectra of these acceptors were measured in the temperature range of 60-20 °C in chloroform, and the results were reported on pp. 8~9: “Besides, the temperature-dependent UV-vis absorption spectra of these acceptors were also measured in the temperature range of 60-20 °C in chloroform to investigate their aggregation properties in solutions (**Supplementary Fig. 31**). All these acceptors show similar spectral change behavior with their increased and red-shifted maximum absorption peaks from 60 °C to 20 °C, which indicates the similar intermolecular interactions and aggregation behavior of these acceptors in chloroform.”

Supplementary Fig. 31. Variable-temperature UV-vis absorption spectra of (a) YDT, (b) DY, (c) TY, (d) QY and (e) PY-IT in chloroform solutions.

4. Large energy loss is a key factor limiting the PCE of OPV. Therefore, the energy loss of devices based on these acceptors needs to be measured.

Response: Thanks for the reviewer’s suggestion. The energy loss of these acceptors-based devices were measured, and the results were reported on pp. 13~14: “To further investigate the internal mechanism of the varied V_{oc} values for these devices, the ΔE_{loss} measurements were conducted for the corresponding devices. The detailed experiment descriptions are shown in the methods part. According to the energy loss measurement, the ΔE_1 values are 0.265~0.267 eV for these five acceptors-based devices. The ΔE_2 values are 0.078 eV for the YDT based device, 0.062 eV for the DY based device, 0.058 eV for the TY based device and 0.022 eV for the QY based device. The ΔE_3 is non-radiative recombination loss and contributes to the largest part of the total energy loss in these devices. The values of EQE_{EL} for this system determine the ΔE_3 , which were summarized in **Supplementary Table 1** and the corresponding curves were exhibited in **Supplementary Fig. 34**. For the acceptors with determined structure, the EQE_{EL} of the devices based on these acceptors gradually decreases with the increased molecular size, leading to the increased non-radiative recombination loss of 0.15, 0.19, 0.202 and

0.253 eV for YDT, DY, TY and QY-based devices, respectively. The enlarged ΔE_3 in the corresponding devices may be due to the increased energetic disorder (**Supplementary Fig. 35**), which may originate from the increased molecular size and lack of halogen substitution in the terminal. Thus, the above energy loss components result in the gradually increased ΔE_{loss} from 0.494 eV for YDT-based device, to 0.519 eV for DY-based device, to 0.526 eV for TY-based device and to 0.540 eV for QY-based device. Compared to the SMA and GMAs, PY-IT with uncertain structure and wide molecular weight distribution exhibits an unusual change in energy loss of its device. This results in a similar ΔE_{loss} of 0.536 eV for the PY-IT-based device compared to that of QY-based device (0.540 eV) although PY-IT may possess larger molecular size, which is probably due to the different interface morphology characteristics between PSMA and GMA-based blends⁴¹. Therefore, the varied ΔE_{loss} of these acceptors-based devices and the progressively down-shifted E_{LUMO} from YDT to DY, TY, QY and PY-IT cooperatively result in the slightly decreased V_{oc} from the YDT based device to the DY, TY, QY and PY-IT based devices.” Meanwhile we added a sentence on p. 13 line 5: “The slightly decreased V_{oc} for the OSCs based on GMAs and PSMA could result from the downshifted E_{LUMO} of the acceptors and the varied ΔE_{loss} of the corresponding devices. The ΔE_{loss} of the YDT, DY, TY, QY and PY-IT based devices will be discussed later.”

In addition, we added a section of “**Energy loss measurements.**” in the methods part on p. 29: “**Energy loss measurements.** Fourier-transform photocurrent spectroscopy external quantum efficiency (FTPS-EQE) was measured by using an integrated system (PECT600, Enlitech). External electroluminescence quantum efficiency (EQE_{EL}) measurements were performed by applying external voltage/current sources through the devices (REPS, Enlitech). The total ΔE_{loss} is determined by the optical gap E_g and V_{oc} ($\Delta E_{\text{loss}} = E_g - qV_{\text{oc}}$), which can be divided into three parts according to the Shockley-Queisser (SQ) limit: $\Delta E_{\text{loss}} = \Delta E_1 + \Delta E_2 + \Delta E_3$ ⁵³. The ΔE_1 is the unavoidable radiative recombination losses above the bandgap and is decided only by the E_g^{PV} of the absorber and temperature: $\Delta E_1 = E_g - qV_{\text{oc}}^{\text{SQ}}$, $V_{\text{oc}}^{\text{SQ}}$ is the maximum V_{oc}

in the SQ limit. The ΔE_2 stands for the radiative recombination losses below the bandgap and is ascribed to non-step function like absorption or EQE of the real-world devices: $\Delta E_2 = qV_{oc}^{SQ} - qV_{oc}^{rad}$, V_{oc}^{rad} is the V_{oc} where only radiative recombination occurs. The ΔE_3 is non-radiative recombination loss: $\Delta E_3 = -k_B T \ln(EQE_{EL})$, EQE_{EL} is the radiative quantum efficiency of the device when charge carriers are injected into the device in the dark.” (The added references are: [41] Ji, Y., Xu, L., Hao, X.&Gao, K. Energy Loss in Organic Solar Cells: Mechanisms, Strategies, and Prospects. *Sol. RRL* **4**, 2000130 (2020); [53] Wang, Y., *et al.* Optical Gaps of Organic Solar Cells as a Reference for Comparing Voltage Losses. *Adv. Energy Mater.* **8**, 1801352 (2018)).

Supplementary Fig. 34. (a) The electroluminescence quantum efficiency at different injected currents, (b) the Fourier-transform photocurrent spectroscopy and (c) the energy loss for the optimized OSCs based on YDT, DY, TY, QY and PY-IT, respectively.

Supplementary Fig. 35. FTPS-EQE of the OSCs based on (a) PM6:YDT, (b) PM6:DY,

(c) PM6:TY, (d) PM6:QY and (e) PM6:PY-IT at the absorption onset.

Supplementary Table 1. Energy loss of the OSCs based on PM6:acceptors.

Devices	$E_g^{PV\ c)}$ (eV)	EQE_{EL}	qV_{oc}^S Q (eV)	qV_{oc}^{ra} d (eV)	ΔE_{loss} (eV)	ΔE_1 (eV)	ΔE_2 (eV)	ΔE_3 (eV)	V_{oc}^C al (V)
PM6:YD	1.482	3.08×10^{-3}	1.216	1.138	0.494	0.266	0.07	0.15	0.98
T		3					8		8
PM6:DY	1.481	6.37×10^{-4}	1.214	1.152	0.519	0.267	0.06	0.19	0.96
		4					2		2
PM6:TY	1.475	4.10×10^{-4}	1.209	1.151	0.526	0.266	0.05	0.20	0.94
		4					8	2	9
PM6:QY	1.476	5.78×10^{-5}	1.211	1.189	0.540	0.265	0.02	0.25	0.93
		5					2	3	6
PM6:PY-IT	1.465	5.47×10^{-4}	1.200	1.123	0.536	0.266	0.07	0.19	0.92
		4					6	4	9

5. The photostability and thermal stability of these devices should be measured.

Response: Thanks for the reviewer's suggestion. The photostability and thermal stability of these devices were investigated and the results were reported on pp. 23~24, as mentioned above in pp. 3~5 in the response letter to reviewers.

Response to Reviewer #3:

In the submitted manuscript entitled "Precise synthesis and photovoltaic properties of giant molecule acceptors", a stepwise precise synthesis method was developed to synthesize giant molecular acceptors (GMAs) with three or four acceptor subunits. This work provides a feasible way to diversify the GMAs for the following study. Furthermore, the impact of the structural transition from small molecules to GMAs on

their structure-property relationship was carefully investigated. The development of GMAs may provide an alternative and promising approach to advance the practical application of organic solar cells. Overall, this work represents the research interests of the community and deserves to be published in Nature Communications after addressing the following comments.

1. These acceptor-based devices show a trend of decreased Voc from YDT-based devices to DY, TY, and QY-based devices and then to PY-IT-based devices. The author mentioned that “The slightly decreased Voc could result from the downshifted ELUMO of the corresponding acceptors.” Energy loss also plays an important role in affecting the Voc of the corresponding device. The authors should quantify the energy losses in the corresponding devices.

Response: Thanks for the reviewer’s suggestion. The energy loss of these acceptor-based devices was measured and exhibited in **Supplementary Fig. 34** and **Supplementary Table 1**. And we revised the explanation of the V_{oc} decrease on p. 13: “The slightly decreased V_{oc} for the OSCs based on GMAs and PSMA could result from the downshifted E_{LUMO} of the acceptors and the varied ΔE_{loss} of the corresponding devices. The ΔE_{loss} of the YDT, DY, TY, QY and PY-IT based devices will be discussed later.” We also added a paragraph to report the results of ΔE_{loss} measurements on pp. 13-14: “To further investigate the internal mechanism of the varied V_{oc} values for these devices, the ΔE_{loss} measurements were conducted for the corresponding devices. ...

2. From solutions to films, the maximum absorption peaks of these molecules present different degrees of redshift, which was explained by the different molecular aggregation characteristics of these acceptors. The molecular preaggregation behaviors of these acceptors in solutions may also affect their absorption properties. The authors are suggested to investigate the molecular preaggregation features in different solvents.

Response: Thanks for the reviewer’s suggestion. The molecular preaggregation

behavior of these acceptors in solutions were investigated by the temperature-dependent UV-vis absorption spectra of these acceptors in the temperature range of 60-20 °C in chloroform, as shown in **Supplementary Fig. 31**. The results were reported on pp. 8-9: “**Besides, the temperature-dependent UV-vis absorption spectra of these acceptors were also measured in the temperature range of 60-20 °C in chloroform to investigate their aggregation properties in solutions (Supplementary Fig. 31). All these acceptors show similar spectral change behavior with their increased and red-shifted maximum absorption peaks from 60 °C to 20 °C, which indicates the similar intermolecular interactions and aggregation behavior of these acceptors in chloroform.**”

3. In the fs-TA measurement, the charge recombination becomes slower with the increase of the SMA units from YDT to PY-IT, which is not matched with the charge carrier recombination behaviors of these OSCs, as investigated by exploring the dependence of J_{sc} and V_{oc} on the illumination light intensity (P_{light}). Please provide more evidence or explanation.

Response: Thanks for the reviewer’s suggestion. The fs-TA measurement is carried out on a bare active layer without electrode, which is different from the measurement of the dependence of J_{sc} and V_{oc} on the illumination light intensity (P_{light}) with a completed photovoltaic device. In the fs-TA measurements, subsequent charge extraction from the active layer to the electrode does not occur in the bare active layer and the charge recombination occurs within the active layer. While the charge recombination behavior measured from the dependence of J_{sc} and V_{oc} on the illumination light intensity (P_{light}) refer to the whole photovoltaic device, which is more directly related to the corresponding device performance. Therefore, there is some difference between the fs-TA measurement results and the behavior of dependence of J_{sc} and V_{oc} on P_{light} , although the main conclusion on the charge recombination is consistent.

4. From SMA to GMA and then to PSMA, the gradually increased number of SMA subunits in these acceptors contributes to their gradually enlarged electron

delocalization structure, which is beneficial for the corresponding charge transport. The authors are suggested to provide comprehend discussion and explain why the TY-based device shows the highest electron mobility among all these acceptors-based devices.

Response: Thanks for the reviewer’s suggestion. The charge transport characteristics measured by the SCLC method and the photo-CELIV are the sum of the intramolecular charge transport and intermolecular charge transport. The increased molecular conjugation length will be beneficial to the intramolecular charge transport, but too large molecular size may hinder the orderly packing of molecules in the film, thereby affecting the intermolecular charge transport ability of the materials. The intermolecular packing properties of these acceptors based neat and blend films have been explored by the GIWAXS measurements. The most ordered molecular packing of TY in neat and blend films and the relatively large electron delocalization structure contribute to the highest electron mobility of TY based device among all these acceptors-based devices. We give the explanations on p. 21: “**These results suggest that the TY-based blend film possesses higher volume fraction of ordered molecular packing, which contributes to the improved charge transport and the better device performance of the TY-based OSCs.**”

5. The authors are suggested to discuss more about the exciton lifetime. For instance, the shorter exciton lifetime vs. the longer exciton diffusion length.

Response: Thanks for the reviewer’s suggestion. The exciton lifetime τ presented in Table 1 refers to the intrinsic exciton lifetime extracted from the dilute acceptor samples. The corresponding description is added on P. 12: “ **τ is the intrinsic exciton lifetime extracted from dilute acceptor solutions**”. The exciton diffusion length L_D is calculated by the equation: $L_D = \sqrt{D\tau}$, which is related both to the intrinsic exciton lifetime τ and the exciton diffusion coefficient D . The exciton diffusion coefficient D is influenced by annihilation rate constant γ (γ is negatively correlated with the exciton lifetime in acceptor film) and annihilation radius R , according to: $D = \frac{\gamma}{8\pi R}$, where R is the annihilation radius from the d_{100} spacing in GIWAXS data. Thus, the shorter exciton

lifetime in acceptor film results in the larger annihilation rate constant γ , which helps to enhance the exciton diffusion coefficient D and further the longer exciton diffusion length L_D . Overall, according the equation, the larger intrinsic exciton lifetime τ and annihilation rate constant γ , the longer exciton diffusion length L_D .

Best regards!

Yongfang Li

Institute of Chemistry

Chinese Academy of Sciences

Beijing 100190, China.

REVIEWERS' COMMENTS

Reviewer #1 (Remarks to the Author):

There are no technical flaws with the paper. The production of the paper is of high quality. The question is on novelty for publication in Nature. As the authors have pointed out in their response letter, many works exist on 'Giant Molecular Acceptors". Personally, I see the compounds as new, not novel, and find it hard to see a major innovation compared to published works by others and the same author. All these molecules are very similar and improved pathways towards making OPVs a reality not clear. But there is no doubt that the OPV community will benefit from the work. At this stage this should be an editor's decision.

Reviewer #2 (Remarks to the Author):

The revision and replies are fine now.

Reviewer #3 (Remarks to the Author):

The authors have majorly revised the manuscript and addressed all the comments. I recommend its publication in Nature Communications.